# Reliable water vapour isotopic composition measurements at low humidity using frequency stabilised cavity ring down spectroscopy

Mathieu Casado[1], Amaelle Landais[1], Tim Stoltmann[2], Justin Chaillot[1], Mathieu Daëron[1], Fréderic Prié[1], Baptiste Bordet[2], Samir Kassi[2]

[1] Laboratoire des Sciences du Climat et de l'Environnement, CEA–CNRS–UVSQ–Paris-Saclay–IPSL, Gif-sur-Yvette, France
[2] LiPHY, Laboratoire Interdisciplinaire de Physique, Université Grenoble Alpes / CNRS, Grenoble, France

*Correspondence to*: Mathieu Casado (mathieu.casado@gmail.com)

**Abstract.** *In-situ* measurements of water vapour isotopic composition in Polar Regions has provided needed constrains of post-deposition processes involved in the archiving of the climatic signal in ice core records. During polar winter, the temperatures, and thus the specific humidity, are so low that current commercial techniques are not able to measure the vapour isotopic composition with enough precision. Here, we make use of new developments in infrared spectroscopy and combine an optical feedback frequency stabilised laser source (OFFS technique) using a V-shaped optical cavity (VCOF) and a high-
finesse cavity ring down cavity (CRDS) to increase the signal to noise ratio while measuring absorption transitions of water isotopes. We present a laboratory infrared spectrometer leveraging on all these techniques dedicated to measure water vapour isotopic composition at low humidity levels. At 400 ppmv, the instrument demonstrates a precision of 0.01 and 0.1‰ in $\delta^{18}O$ and d-excess, respectively, for an integration time of 2 minutes. This set up yields an isotopic composition precision below 1‰ at water mixing ratios down to 4 ppmv, which suggest an extrapolated precision in $\delta^{18}O$ of 1.5‰ at 1 ppmv. Indeed, thanks
to the stabilisation of the laser by the VCOF, the instrument exhibits extremely low drift and very high signal to noise ratio. The instrument is not hindered by a large isotope-humidity response which at low humidity can create extensive biases on commercial instruments.

## 1 Introduction

Water isotopic composition is commonly used as an atmospheric tracer (Galewsky et al., 2016) or in paleoclimate studies
(Casado, 2018; Jouzel and Masson-Delmotte, 2010). Indeed, water stable isotopes concentrations are modified throughout the hydrological cycle (Dansgaard, 1964), in particular at each phase transitions (Craig and Gordon, 1965; Majoube, 1971; Merlivat and Nief, 1967), heavier isotopes are preferentially found in the condensed phase rather than the vapour (Jouzel, 2010). This property is paramount to the isotopic paleothermometer (Dansgaard, 1964; Lorius et al., 1969) which links the variations of isotopic composition in an ice core record to past temperatures (EPICA, 2004; North Greenland Ice Core Project
members, 2004). While the isotopic paleothermometer is generally admitted to be a reliable paleoclimate proxy (Jouzel and Masson-Delmotte, 2010), in low accumulation regions, complex post-deposition processes (Casado et al., 2018; Ekaykin et

al., 2002) can modify the recorded signal (Casado et al., 2021; Steen-Larsen et al., 2014) and limit the interpretation as a past temperature record (Casado et al., 2020; Laepple et al., 2018).

Studying water vapour isotopic composition in polar regions (Casado et al., 2016; Steen-Larsen et al., 2013) is key to a comprehensive understanding of the processes affecting water isotopes in cold, dry environment, usually characterised by low accumulation (Berkelhammer et al., 2016; Bonne et al., 2019; Bréant et al., 2019; Casado et al., 2018; Ritter et al., 2016). However, in the low accumulation regions of the poles, vapour monitoring of isotopic composition is sparse (Wei et al., 2019) and for the coldest sites limited to summer. Indeed, these measurements are mostly based on Picarro commercial instruments

(Steig et al., 2014) for which the precision drops dramatically for humidity levels below 100 ppmv (Leroy-Dos Santos et al., 2021). Throughout this manuscript, we will use Picarro instruments as a benchmark for commercial instruments considering how ubiquitous they are in water vapour isotopic composition monitoring. In general, the amount of sublimation and condensation is poorly known during the winter months (Genthon et al., 2016), and there is no instrument able to monitor the atmospheric vapour isotopic composition due to the extremely low humidities (below 100 ppmv) (Casado et al., 2016; Ritter

et al., 2016). This is despite attempts for new generation of infrared spectrometers which were able in laboratory environments to reach humidity as low as 20 ppmv (Landsberg et al., 2014). At the site of Dome C, where the longest ice core has been drilled, humidity levels below 100 ppmv are found 79 % of the year, and below 20 ppmv, 56 % of the year (Genthon et al., 2022), yielding an urgent need for vapour isotopic composition monitoring able to measure in conditions as dry as 1 ppmv.

Applications of infrared spectroscopy techniques to water isotopic monitoring is dominated by two techniques: Optical-Feedback Cavity Enhanced Absorption Spectroscopy (OF-CEAS) (Landsberg, 2014) and Cavity Ring Down Spectroscopy (CRDS) (Romanini et al., 1997; Steig et al., 2014). The advantage of the former is to use optical feedback (Laurent et al., 1989) to stabilise and refine the laser source and measure the absorption features associated to molecular transitions in a high finesse resonating cavity (Morville et al., 2005). The latter on the other hand takes advantage of the extreme sensitivity of the

CRDS technique (Čermák et al., 2018) to perform very precise absorption spectroscopy.

Here, building up on recent efforts to combine the advantages of both techniques (Burkart and Kassi, 2015; Chaillot et al., 2023; Stoltmann et al., 2017), we present a new generation of infrared spectrometers able to measure water isotopic composition at extremely low humidities (less than 1 ppmv, or equivalent to a water pressure of 0.06 Pa). These conditions are

indeed commonly found in central Antarctica during the long polar winters where temperature can reach -90 °C, but are also potentially found in the stratosphere or on other planets. This instrument, called VCOF-CRDS (V-shaped Cavity Optical Feedback – Cavity Ring Down Spectroscopy), is able to reach precision of 0.01 and 0.1‰ for monitoring vapour $\delta^{18}O$ and d-excess (second order parameter d-exc = $\delta D - 8\ \delta^{18}O$), respectively, and should reach satisfactory performances at humidity around 1 ppmv. Overall, the instrument achieves a precision roughly 20 times better than available commercial instruments,

as well as a stability 20 times longer. An ingenious auto-referencing of the instruments also limits the impact of long term drift

of the laser and reduces the need for drift correction through regular calibration. While the current prototype cannot be deployed to the field due to its relative bulkiness ($1m^3$), weight, and fragility, this manuscript intends as providing a proof of concept for the application of this technology for monitoring of water vapour isotopic composition at humidity levels down to 1 ppmv.

## 2. Methods and instrumental setup

### 2.1. Description of the instrument

The instrument is based on Optical Feedback Frequency Stabilisation (OFFS) Cavity Ring Down Spectroscopy (CRDS), a technique developed for $CO_2$ isotope monitoring (Burkart et al., 2014; Stoltmann et al., 2017) and transferred here to water isotopic composition measurement. It includes three modules: a laser source, a frequency scanner, and a gas analyser (see Figure 1).


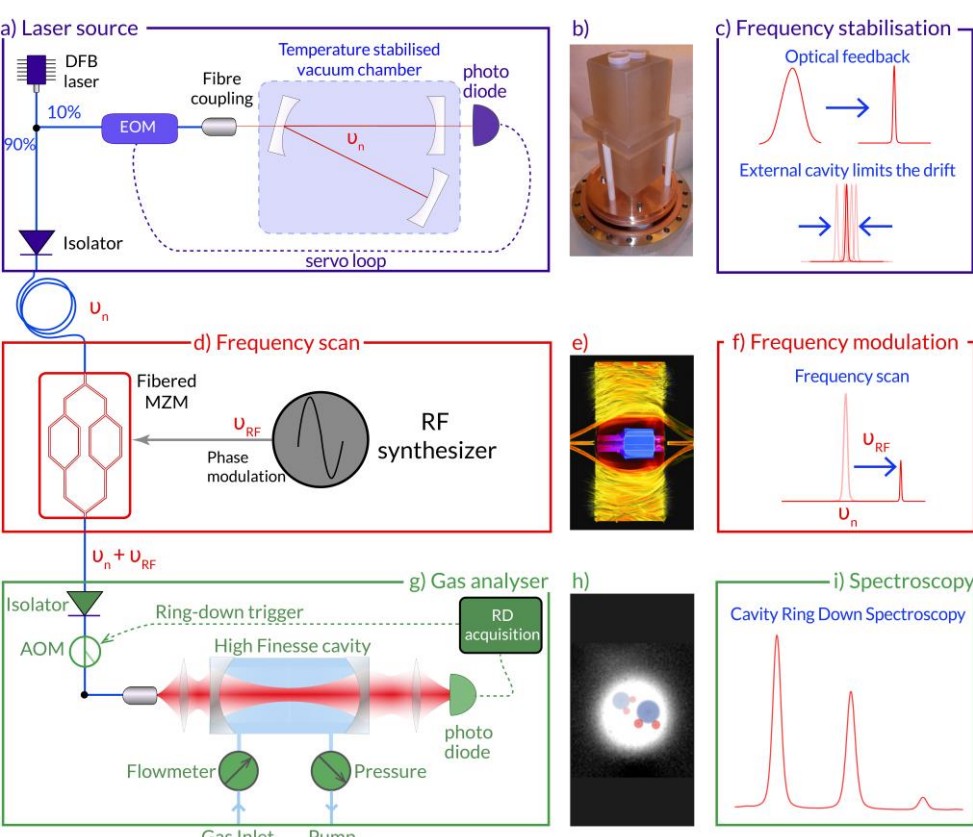

***Figure 1****: Experimental setup schematics: a) the laser source, b) a picture of the monobloc zerodur V-shape stabilisation cavity, c) the impact of the optical cavity on the linewidth and the drift of the laser source; d)frequency scanner through a Mach-Zehnder Modulator (MZM), e) AI-generated representation of a MZM, f) illustration of the sideband generation through*

 *the MZM; g) the CRDS cavity used to analyse the gas, h) silhouette of two water molecules drawn on top of an IR picture of the mode of the cavity, and i) a schematic spectrum for the considered transition line (see Fig. 2).*

The light source is primarily composed of a fibered Distributed FeedBack laser diode (DFB, Eblana Photonics) stabilised by optical feedback (Fig. 1a) in a monoblock Zerodur V-shaped cavity (Fig. 1b). The optical power of the laser diode is approximately 20mW, by operating it at a temperature of 20.997°C with a 122mA current. The mirrors (Layertec) have a reflectivity of 0.99992, leading to a cavity with a finesse of 131000. The frequency stabilisation leads to the narrowing of the linewidth of the crude DFB diode from a few MHz to less than 70 Hz (Fig. 1c), with a drift below 2 Hz s$^{-1}$ (Casado et al., 2022). The long term drift in particular drops below 1 Hz s$^{-1}$, and is dominated by the aging of the zerodur glass (Jobert et al., 2022) providing from a spectroscopy point of view long term stability of the instrument, and thus, reducing the need for calibration to compensate the drift. The entire laser source set-up is regulated at 26°C by heating elements and PT1000 temperature sensors as described in (Casado et al., 2022; Jobert et al., 2022).

The frequency scanner relies on a fibered Mach-Zehnder Modulator (MZM, Fig. 1d-e) (Izutsu et al., 1981) which enables tuning of the frequency of the light source while preserving its spectral purity (Burkart et al., 2013). We apply an electrical radio-frequency signal to shift the sidebands generated by the MZM (Fig 1f), and set up the polarisation so only a single sideband is preserved while the other and the carrier are destroyed by negative interferences. The remaining power is around 0.9 mW, and we use a fibered optical amplifier to increase the power to 11 mW.

The light is then injected into a second cavity which is used for monitoring the isotopic composition of the water vapour based on Cavity Ring Down Spectroscopy (CRDS, Fig 1g). The CRDS cavity consist of a 48 cm long copper cylinder, with a diameter of 7 cm in which a hole of 0.8 cm has been drilled over the full length. The reflectivity of the mirrors (Layertec) is 0.99995, leading to a finesse of 170 000. The length of the CRDS cavity can be adjusted by a piezo ceramic actuator which moves one of the mirrors. This is used to adjust the length of the CRDS cavity, and thus its resonance frequency, to match the frequency of the light injected. As mentioned above, the power injected to the cavity is amplified to 11 mW to increase the SNR on the photodiode while ensuring that saturation is not affecting the absorption profile of the gas inside the cavity (Kassi et al., 2018). An optical power of roughly 1mW exits the CRDS cavity and is collimated on a Hamamatsu photodiode. The gas is injected in the cavity using pressure and flow controllers (Bronkhorst). The cavity is maintained at a pressure of 35 mbar (±0.01mbar), and the flow at 25 sccm to limit turbulences within the air flow. The entire gas analyser (Fig. 1g) is stabilised using a Peltier device (Supercool) at a temperature of 28°C. The CRDS cavity is stabilised using a heating wire and PT1000 temperature sensors at 29.000°C.

## 2.2. Spectroscopic analysis

The instrument is set such that the laser source produces light in an extremely stable manner at exactly 7199.7 cm$^{-1}$, the frequency of which is then shifted using solely the MZM. The instrument can then be used in two ways: slow full spectrum mode and high pace measurement mode.

In full spectrum mode, the instrument has a high spectral resolution and can be used to realise spectroscopy studies (Kassi et al., 2018), with a sensitivity of $10^{-12}\ cm^{-1}$ after 60 s. At high pace measurement mode, i.e. using the instrument as a tool to monitor water vapour isotopic composition, the pace of the measurement is key in order to avoid the gas concentration changing during the course of a spectrum. Instead of measuring spectra including a large number of datapoint, in this case, we

opt out for a large number of spectra with few datapoints at the most relevant frequencies to characterise the absorption lines. This is done by "jumping" exactly one Free Spectral Range (FSR) of the CRDS cavity using the MZM. With this approach, the spectral datapoints are spaced by multiples of the FSR (around 297 MHz), but relocking is instantaneous. By preventing any changes in gas composition throughout the duration of the scan (0.3 s), this technique improves on higher resolution scans (up to 1 minute), which, while potentially more precise, are often far slower. Fast variations of humidity level during a scan

create large uncertainty in the fitting procedure which lead to additional fluctuations on the isotopic composition which are not averaged due to their auto-correlated features. The high pace scans are associated with higher instantaneous noise with little auto-correlation which can be averaged out rapidly. As a result, when the high pace scans are averaged out to the acquisition time of the slow pace measurements, the precision of the average high pace scans is better than the precision of single slow pace scans.

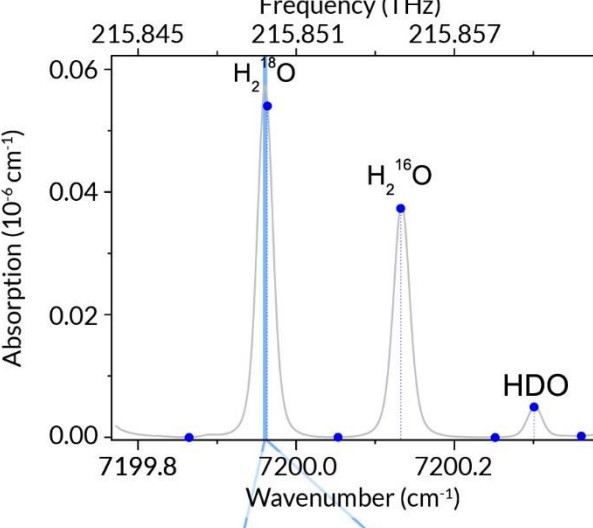

a) Measurement mode

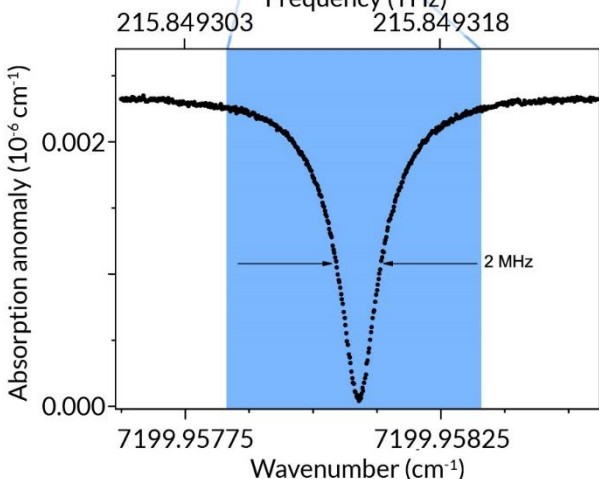

b) Frequency referencing

*Figure 2: Absorption spectra realised by the instrument: a) in high pace measurement mode at 400 ppmv, 35 mbar (blue dots), compared to a slow full spectrum(grey points), and illustration of the area scanned for frequency referencing (blue shaded area); b) in frequency referencing mode, at 0.1 mbar: feature of a Lamb dip saturated absorption pattern used to evaluate the drift of the instrument when deployed in the field where comb assisted spectroscopy is not possible.*

In the high pace measurement mode, the spectra are composed of 7 datapoints (one at the top of each transition, and four for the baseline, Fig. 2a) and are realised in approximately 0.3 s when averaging together 10 ring downs per spectral point. We fitted a slow pace/high precision spectrum using a speed-dependent Nelkin-Ghatak profile (SDNGP) (Long et al., 2011), similarly to the approach used to measure $CO_2$ isotopic composition (Stoltmann et al., 2017) to fix pressure and temperature dependant parameters. As temperature and pressure are constant, these parameters can be fixed which reduce the number of

free parameters. We then were able to generate a simple, multi-linear, conversion matrix which could link the intensity of each transition in the spectra to the concentration of each isotope.

## 2.3. Frequency referencing

Even though the drift of the laser source is extremely limited (below 2 Hz s$^{-1}$), at the scale of several hours, this can lead to significant deviations of the frequency, which lead to drift of the measured isotopic composition. This issue is general for infrared spectrometers and is usually tackled by post-correction of empirically established drift of the measurement of the isotopic composition itself (Leroy-Dos Santos et al., 2021). For previous VCOF setups, the frequency of the VCOF has been locked on an optical frequency comb which is itself locked on the GPS signal (Gotti et al., 2018). Here, we propose instead to measure a Lamb dip saturated absorption feature to counteract the drift of the laser source. Indeed, at low pressure, the number of molecules can be so low that a *dip* feature appears at the centre of saturated absorption transitions (Fig. 2b) (Burkart et al., 2015; Kassi et al., 2018). These features are characterised by very small linewidth (below 20 kHz if the pressure is low enough), and their frequency, which remains constant for constant conditions of pressure, can be used as a frequency reference.

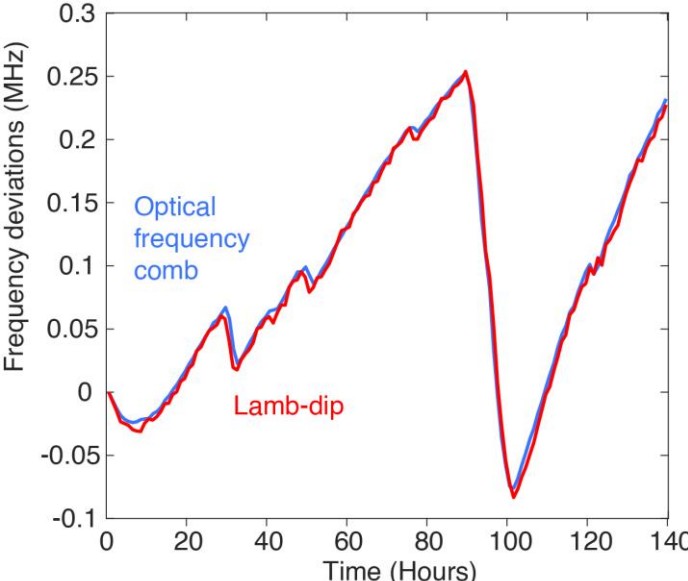

*Figure 3: Frequency deviation of the laser source as measured from the beatnote of the laser with an optical frequency comb (blue,) compared to measurement from the centre of a lamb-dip saturated absorption feature (red)*

Here, we make use of this property to evaluate the frequency drift of the laser source by scanning every hour the Lamb dip feature associated with the H$_2^{18}$O transition at 7199.96cm$^{-1}$. To do so, we decrease the pressure inside the cavity down to 0.1 mBar, let the cavity stabilise to these new experimental conditions for 2 minutes, and then measure Lamb dip feature across 4

MHz at extremely high resolution (30 kHz) for 7 minutes. This method provides with frequency measurements of the Lamb dip feature centre with a precision of 2.5 kHz in a single scan (Kassi et al., 2018). During a period of 140 hours, we measure the frequency deviation from the Lamb dip method, which reproduces exactly the ones estimated from the measurement of a beatnote with an optical frequency comb locked on a GPS (Burkart et al., 2014) (Fig. 3), with a correlation between the two times series of $r^2 = 0.995$ ($p < 0.05$). The pressure is then increased back to 35 mbar, within 30 seconds. This leads to an biased isotopic composition measurement for another 3 minutes, after which, we can measure again the vapour isotopic composition. The overall duty-cycle here was around 80%, including 13 minutes every hour during which the instrument was not able to monitor isotopic composition. Improvements on the VCOF laser source as recommended in (Jobert et al., 2022) could limit the need for such self-referencing cycles. Here, we used this measurement of the deviation of the frequency of the laser source as a self-referencing method which can be applied every hour to ensure that the deviation of the laser source frequency remains smaller than 10 kHz.

## 2.4 Generation of water vapour standards

We used the water vapour generator instrument described in (Leroy-Dos Santos et al., 2021) to generate stable humidity levels to evaluate the response of the infrared spectrometer to varying humidity levels. We used dry, synthetic air bottles with less than 3 ppmv of water to supply the gas to the instrument. We used laboratory standards of water with isotopic composition varying from 0 to -60 ‰ to supply with water of known isotopic composition. The water vapour generator shows relatively good performances down to humidity around 20 ppmv where outgassing from the instrument itself limits its performance. As a result, we used the water vapour generator to evaluate the precision as a function of humidity (section 3.1) and the humidity response of the instrument (section 3.2) down to 25 ppmv.

To obtain an evaluation of the precision of the instrument below these humidity levels, we connected a dry air bottle directly to the instrument and regulated the produced humidity levels from outgassing of the tube by changing the flux of dry air to an exhaust connected to a sonic nozzle. Using this method, humidity levels down to 4 ppmv could be reached. In this case, the isotopic composition is not known so it cannot be used to evaluate the accuracy of the instrument, only how stable the instrument is capable to measure relatively stable isotopic composition of outgassing water from the tube walls. As the temperature of the tube and the dry air canister were not regulated nor monitored, the resulting water vapour was relatively variable (variations around 10% of the produced humidity level), which lead to potential variability in the isotopic composition, leading itself to relatively high Allan standard deviations not necessarily linked with poor performance of the instrument.

## 3. Results

We discuss the performances of the instrument with first, the precision and the long-term drift of the instrument; second, the accuracy of the instrument, including the humidity-to-isotope and isotope-to-isotope relationships; and finally by highlighting the impacts of the frequency auto-referencing on the performances of the instrument.

### 3.1. Precision and drift

The VCOF-CRDS instrument precision has been evaluated by measuring water standards at various humidity levels generated
by the calibration device described in Leroy- Dos Santos et al,. (2021). To evaluate the drift in measured isotopic composition associated with the technique directly, and not be impacted by the potential drift of the laser source, we used an optical frequency comb referenced to a GPS system to actively correct the frequency drift of the laser source. The impact of drift and the mitigation by the frequency auto-referencing scheme are detailed in section 3.3. We generated repeated injections of the same water samples at the same conditions to generate water concentrations as stable as possible that is measured by the
instrument continuously. Typically, at 400 ppmv, this resulted in successive stable humidity levels lasting up to 10 hours, followed by a drop of the humidity to refill the syringe (Fig. A1). The instrument then needs up to 1 hour to generate stable water again. This leads to time series of stable vapour content that can last several days but include gaps. We modified the traditional calculation of Allan variance to take gaps into account to calculate the precision as well as the drift of the instrument in a range of humidity from 20 to 1500 ppmv despite the gaps during the refill of the vapour generator.

To evaluate both the short and long term stability of the signal, we monitored a constant humidity level of 400 ppmv for 7 days (Fig. 4a)). At the temporal resolution of the instrument (3 Hz), the precision is roughly 0.2 ‰ in $\delta^{18}O$ at a humidity of 400 ppmv, and below 0.1 ‰ at 1 Hz. In these conditions, a Picarro L2130i in HDO mode only reaches precision of 2 ‰ at 1 Hz (Casado et al., 2016). The Allan standard deviation follows the behaviour of a white noise (characterised by a $1/\sqrt{N}$ decrease)
until approximately 150 seconds when a limit value of roughly 0.01 ‰ in $\delta^{18}O$ variations is reached. The precision remains at this lowest value for duration longer than 2 days. Similarly, for d-excess, the instantaneous precision at 400 ppmv is roughly 2 ‰, and drops to 0.1 ‰ after 150 seconds (Fig. 5). We chose to showcase $\delta^{18}O$ and d-excess (instead of $\delta D$) because correlated drift between the two first order isotopic compositions (Appendix B) lead to an Allan standard deviation of d-excess characterised with excess noise compared to both the Allan standard deviations of $\delta^{18}O$ and $\delta D$ (Figure B1). The fundamental
flicker noise limit (flat Allan deviation for time larger than 2 minutes) could either be linked with the instable gas generation (variable humidity and pressure) or with the measurement technique itself. At low humidity (25 ppmv), more correlated noise appears, with a decrease of the Allan standard deviation not following the $1/\sqrt{N}$ law. After 1 minute, a precision of 0.2 ‰ in $\delta^{18}O$ is reached and maintained for scales longer than 1 day (we chose not to interpret the drop after 6 hours below 0.1 ‰, as it is only visible in the last two points of the Allan standard deviation). Similarly, the instantaneous precision (1 second) of the
instrument in d-excess at low humidity is very large (around 10 ‰) and slow drops to 2 ‰ around 1 minute.

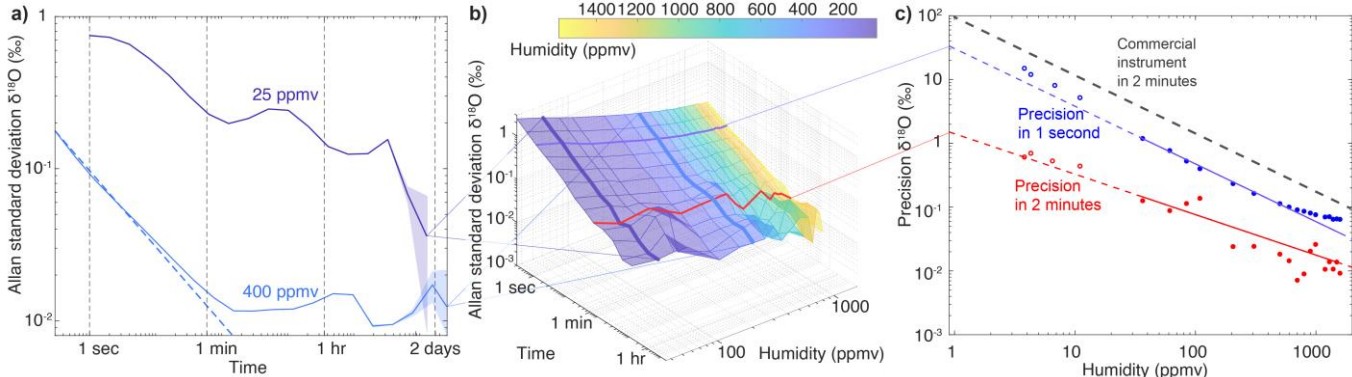

***Figure 4:*** *Allan standard deviation plots of the δ¹⁸O measurements of the VCOF-CRDS instrument stabilised with an optical frequency comb: a) long-term Allan standard deviation plots realised at humidity levels of 25 and 400 ppmv with stable measurements of 7 days, b) 3D plot of the Allan standard deviation for different time scales and humidity levels, and c) evaluation of the precision of the instrument across humidity levels (dots) for measurements average over 1 second (blue), 2 minutes (red), as well as fit with a power law (solid line) and extrapolation from the fit at lower humidities that could not be reached with the calibration device* (Leroy-Dos Santos et al., 2021)*; additional measurements between 3 and 10 ppmv obtained from outgassing tubes (section section 2.4) are included on the graph as upper limits of the precision below 25 ppmv; compared to typical commercial instrument behaviour (dashed grey line, linear approximation of the performances of a Picarro L2140i extracted from Fig. 3 of Leroy-Dos Santos et al, (2021) and of a Picarro L2130i from Casado et al, (2016)).*

The change of precision scales with humidity as shown in Figure 4c, and so at 1 s, the precision of the instrument at 30 ppmv is roughly 2 ‰ in δ¹⁸O (roughly 10 times larger than at 400 ppmv since the humidity is approximately ten times smaller), the precision in δ¹⁸O at 30 ppmv drops to 0.1 ‰ after 800 seconds (Fig. 4b). Reciprocally, at 800 ppmv, the precision in δ¹⁸O is around 0.1 ‰ at 1 s and dropping to 0.005 ‰ after 800 seconds. While we were not able to generate stable moisture flux at 1 ppmv using the humidity generated described in (Leroy-Dos Santos et al., 2021), we extrapolated the precision expected after two minutes at humidity lower than 25 ppmv by fitting the data with a power law and find a precision of 1.5 ‰ at roughly 1 ppmv. We also directly connected the instrument to dry air bottle controlling the humidity from the outgassing of the tubes connecting the bottle to the instrument (see Methods, Section 2.4) and evaluated the precision at 3.8, 4.2, 6.5, and 11 ppmv. Since this method to generate water standard is extremely dependent on temperature variations, it is only useful as an upper bound of the precision of the instrument as the humidity levels are relatively variable (standard deviation of the humidity larger than 10% of the humidity content), and thus, these datapoints were not included in the power law fit. We find precisions ranging from 0.5 to 0.7 ‰ after two minutes (Fig. 4c) which agree relatively well with the power law defined at larger humidity levels.

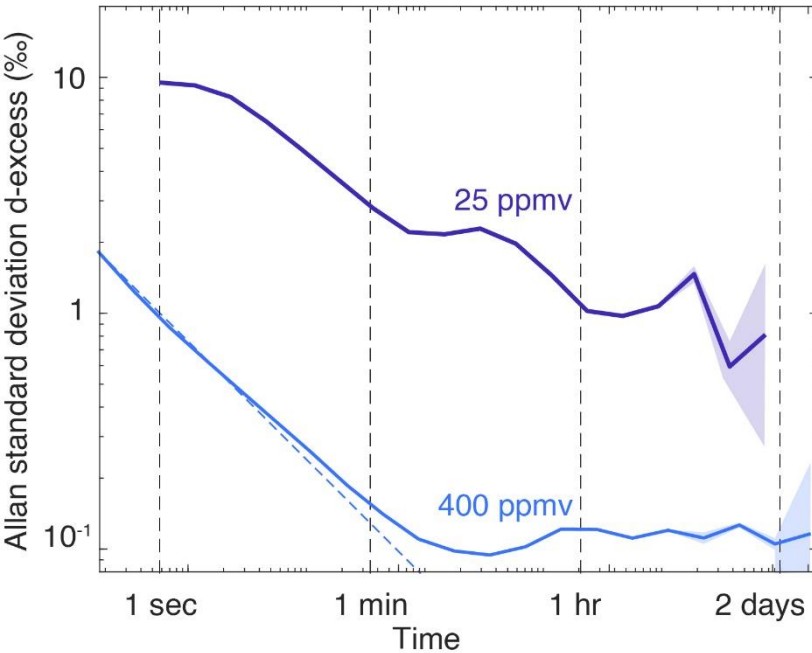

*Figure 5: Same as Figure 4a) for the d-excess: long-term Allan standard deviation plots realised at humidity levels of 25 and 400 ppmv with stable measurements of 7 days.*

**3.2. Accuracy of the instrument**

Infrared spectrometers tend to be biased and require calibration for the dependency of their measurements with respect to
250 change of humidity and isotopic composition itself. To evaluate the humidity response of the VCOF CRDS, we used the water vapour generator to generate different humidity levels and evaluate how the measured isotopic composition deviates at low humidity from the expected value for the given standard. Here, we illustrate the behaviour of the infrared spectrometer for varying humidity levels. We show a nearly flat humidity response for humidity levels above 200 ppmv (Figure 6).

Infrared spectrometers usually have an isotopic response of the measurements to change of humidity levels that needs to be corrected with an appropriate instrument (Leroy-Dos Santos et al., 2021; Weng et al., 2020). Weng et al, (2020) suggested that the humidity response of the measurement of the isotopic composition, in particular the fact that this response is different for different isotopic composition, is linked with spectroscopic effects in Picarro instruments (Grey lines in Fig. 6), rather than background humidity or memory effects, at least at the first order. Here, we show that the improved frequency stabilisation
and the new fit parameters provide a much flatter isotope-humidity response, for both $\delta^{18}O$ and d-excess. The amplitude of the isotopic-humidity calibration changes is indeed one of the largest limitations to interpret vapour isotopic composition at low humidity conditions (Casado et al., 2016; Leroy - Dos Santos et al., 2020). In particular, in Antarctica where humidity falls below 500 ppmv frequently, the correction applied to $\delta^{18}O$ and d-excess (at 100 ppmv, up to 15 and 100 ‰, respectively) can

be one order of magnitude larger than the signal (diurnal cycle around 5 and 10 ‰, respectively at Dome C (Casado et al.,
2016)). Here, the amplitude of the correction remains below 1 and 10 ‰, respectively, for humidities larger than 100 ppmv.

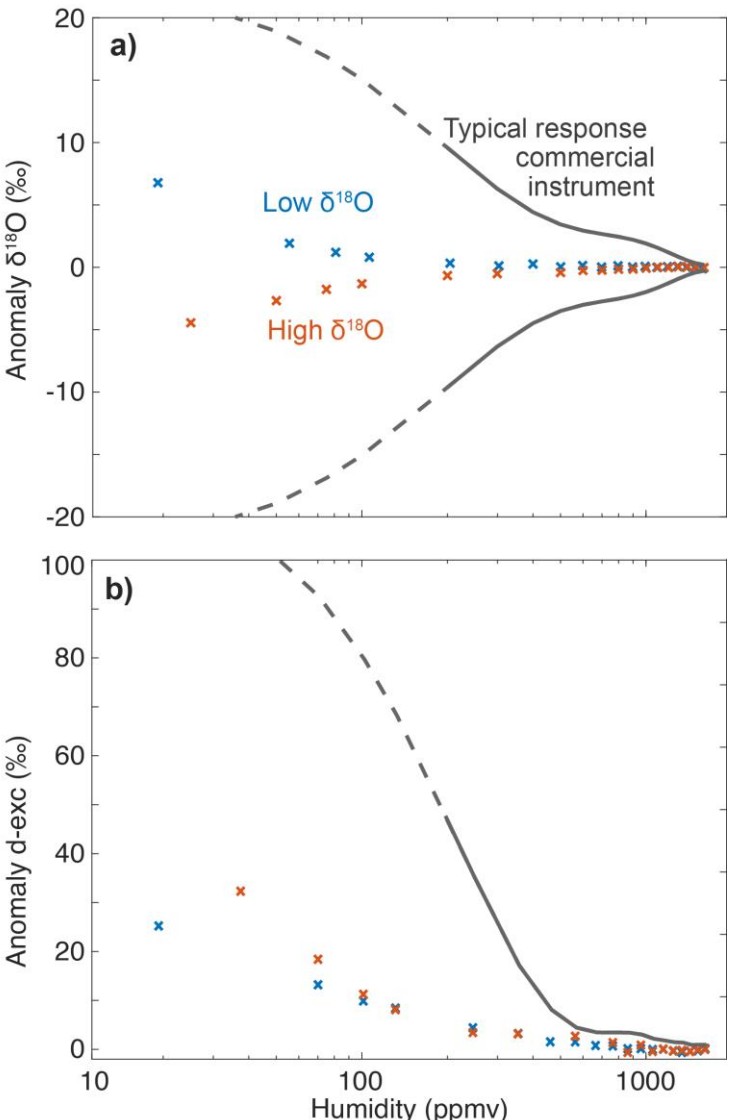

*Figure 6: Isotope-humidity dependence for two isotopic compositions: evaluation of the anomaly of the measurement against the value of the standard of a) δ¹⁸O and b) d-excess for local tap distilled tap water (high δ¹⁸O, approximately of -7.5 ‰, orange) and depleted Antarctic water (low δ¹⁸O, approximately of -50.6 ‰, blue) compared to typical response from a*
*commercial instrument at low humidity (grey lines, extracted from the equations 4 and 5 in Leroy-Dos Santos et al., 2021, based on a Picarro L2140i, full line: humidity range where the regression was made from the observations, dashed line, extrapolation).*

Additionally, for Picarro instruments, the humidity response is very variable in between individual instruments (Weng et al., 2020), and needs to be evaluated for each new analyser, as well as each time an analyser is deployed (Leroy-Dos Santos et al., 2021). The limited humidity response of our setup is therefore extremely valuable but would need to be validated on a second VCOF-CRDS instrument to be confirmed.

Another aspect of the accuracy of isotopic analysers is the linearity of the instrument, or its isotope-isotope response. We evaluated the accuracy of the new instrument on the V-SMOW/V-SLAP scale by measuring six internal standards from our institute ranging from –54 to +0.5‰. We used the water vapour generator (Leroy-Dos Santos et al., 2021) to generate stable humidity levels of 90 minutes and included the average value of the last 15 minutes. As this water vapour generator is not as versatile as an automatic sampling device, injecting water with different isotopic composition (especially across a range of dozens of permil) is cumbersome due to extended memory effects in the humidity generator and it was necessary to wait more than 12 hours to do a new isotopic sample. The measured $\delta^{18}O$ aligns perfectly with the internal standard values ($r^2$ virtually undifferentiable from 1, N = 9), and the residuals of the linear regression have a standard deviation of 0.04‰ (Figure 7).

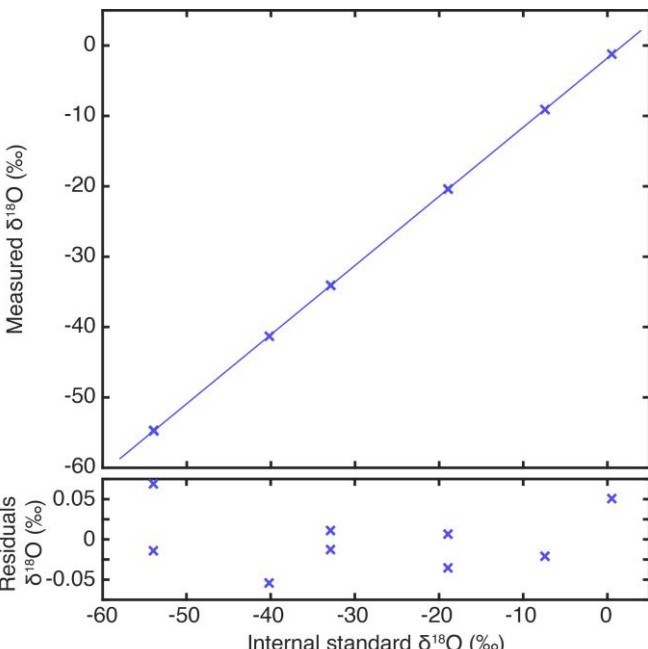

***Figure 7:*** *Isotope-isotope response: measurements of internal standards referenced at average humidity of 800 to 1200 ppmv on the SMOW-SLAP scale and linear regression of the measured values against the standard values, compared to the residues. This demonstrates the linearity of the isotopic measurement of the VCOF-CRDS. Overall, isotopic monitoring via CRDS technique has been demonstrated to be extremely linear, even outside of the range of the isotopic compositions used for calibration (Casado et al., 2016; Steig et al., 2014). In practice, for isotopic composition measurements of ice core samples, it*

would be necessary to use the same sample preparation line (auto-sampler and vaporiser) for samples and calibration. If the residuals had the same type of distribution (standard deviation around 0.04 ‰), to reach an accuracy of 0.01‰ in $\delta^{18}O$, it would be necessary to perform calibration on a smaller range of isotopic composition or to include more measurements of the standards.

### 3.3. Frequency auto-referencing

The main source of uncertainty for the measurement comes from the drift of light emission frequency of the laser source. Indeed, since the frequency scan is only used to reach successive resonance mode of the CRDS cavity, the spectra produced in high pace measurement mode are not necessarily exactly aligned with the absorption lines. For the current cavity, with a FSR of 297 MHz, it was not possible to have datapoints exactly on top of all three scanned transitions, and in particular, the spectral point for the $H_2^{18}O$ transition is relatively far from the transition centre (Fig. 8d).

We evaluated the impact of frequency drift on the measurement uncertainty by artificially adding frequency detuning during the frequency scanning system measuring the same gas in stable conditions. The added detuning of the frequency ranged from -100 to +100 MHz with a 50 MHz resolution, as well as every 1 MHz between -10 and +10 MHz (Fig. 8a, c and e). For the HDO transition where the measurement point was the closest to the transition centre, we observe no impact of the frequency detuning. For the humidity measurement, a small impact is observed (around 0.1 ppmv MHz$^{-1}$) linked to the small distance of the spectroscopic measurement to the centre of the absorption feature. The main impact for our current setup is on $\delta^{18}O$ for which a sensitivity to drift of roughly 1.4 ‰ MHz$^{-1}$ is observed. Considering the current performances of the laser source (1.7 Hz s$^{-1}$, (Casado et al., 2022)), this leads to a drift of the $\delta^{18}O$ of roughly 0.008 ‰ per hour, or up to 0.2 ‰ per day. This is much larger than the performances obtained in Figure 4a) at time scales longer than one hour which suggests that at long time scales, this would be the dominating source of uncertainty.

In order to maintain the performances around 0.01 ‰ for $\delta^{18}O$ for averages longer than two minutes, we need to include frequency referencing every hour. We implemented the frequency auto-referencing mode described in Section 2.3. and measured Lamb dip features every hour. The frequency drift of the laser source measured by this technique was then compensated directly in the frequency modulation by the MZM. Note that no extrapolation of future drift is implemented, so the frequency added on the frequency modulation is a step which is updated every hour when a new Lamb dip is measured. The use of the Lamb dip self-referencing approach enables to actively correct all frequency deviation of the VCOF laser source (Fig. 1a), leading to the performances displayed in Figure 8g).

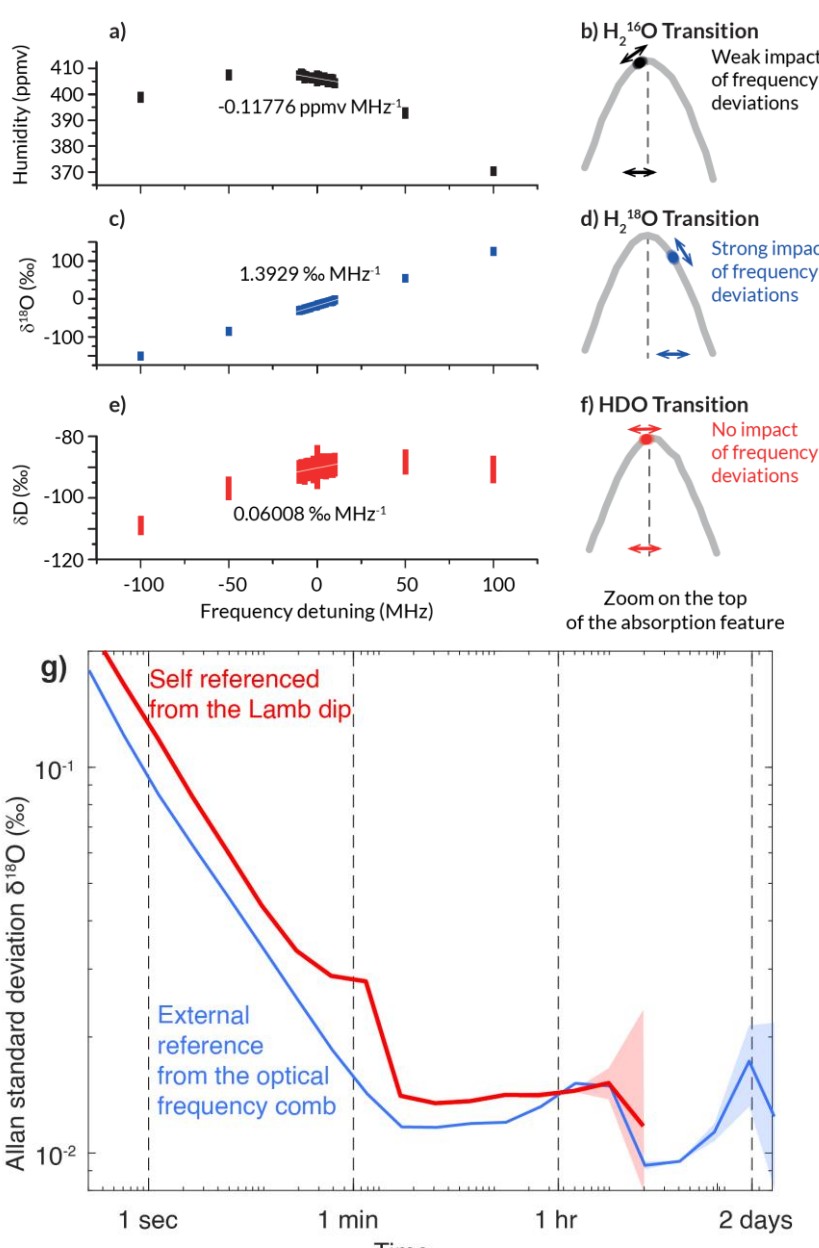

**Figure 8:** *Evaluation of the impact of the frequency drift of the laser source on the measurements of isotopic composition: impact of the frequency detuning on the measured a) humidity, c) $\delta^{18}O$, and e) d-excess with illustration of the position of spectroscopic measurement point compared to their respective transition centre b), d), and f); Allan standard deviation of $\delta^{18}O$ for the instrument being self-referenced (red) and with an external frequency reference (blue) at 400 ppmv.*

We compare the Allan standard deviation of measurement of a constant sample made with this self-referenced mode of the instrument with the Allan standard deviation with the external frequency reference (Fig. 8g). For time scales shorter than one hour, an excess noise is added to the data, probably due to the step approach to correct the drift. An extrapolation from a given number of the previous Lamb dip measurements could potentially on average fix part of this excess noise, but with the risk of over-correction when the environmental conditions shift. Since despite this excess noise, a satisfactory precision can be reached after 5 minutes, this has not been implemented. For time scales longer than one hour, the performances of the instrument with the self-referencing are matching exactly the ones with external frequency referencing. The measurement of δD is not affected as the measurement point is perfectly centred on the transition.

Changing the length of the CRDS cavity so the FSR aligns the frequency of the measurement at the top of all the transitions would mitigate a large part of the drift impact on $\delta^{18}O$ (this is equivalent to finding a FSR equal to the common denominator between the difference of frequency between the three transitions). This solution will be favoured for instrument deployed to the field, and would limit the need to implement frequency auto-referencing to every 6 hours.

## 4. Discussions

Major new study topics arising from the capacity of infrared spectrometers to measure the isotopic composition at low humidities, include water vapour monitoring in very cold field conditions, as well as evaluation of the fractionation coefficients of heavy isotopes with respect to light isotopes (Majoube, 1971).

Current commercial instruments are limited to measure isotopic composition at humidity levels above 100 ppmv (Casado et al., 2016; Leroy-Dos Santos et al., 2021; Ritter et al., 2016). In general, this limits the application of the instruments to locations where the temperature is above -40 °C, and successful deployments has been reported in a large number of Arctic field campaigns (Akers et al., 2020; Bonne et al., 2014; Leroy - Dos Santos et al., 2020; Steen-Larsen et al., 2013), on vessels in Polar regions (Kurita et al., 2015; Thurnherr et al., 2020), and even in coastal areas in Antarctica (Bagheri Dastgerdi et al., 2021; Bréant et al., 2019). In inland Antarctica, some campaign monitored water vapour isotopic composition but were limited to the warmest summer conditions (Casado et al., 2016; Ritter et al., 2016).

Developing infrared spectrometers able to yield satisfactory precision at low humidities has been attempted in the past, making use of the high sensitivity of OFCEAS techniques (Landsberg et al., 2014) but deployment to an Antarctica field station was never successful due to the strong impact of slow drifting fringes which induced significant drift after a couple of hours (Casado, 2016; Landsberg, 2014). The instrument presented here, based on the VCOF-CRDS technique, shows potential to measure at humidity levels two orders of magnitude lower than current capabilities of Picarro instruments (Figure 9). It should be able to have a precision better than 1 ‰ roughly 90 % of the time at the deep interior station Concordia (located at Dome

C) where winter temperatures are usually around -80°C (Genthon et al., 2021). Combining the stability of the frequency stabilised laser source (OFFS and Lamb dip frequency referencing) and the high precision and stability of CRDS technique, the instrument can circumvent all the hurdles that limit the monitoring of water vapour isotopic composition in the coldest conditions, such as found in Antarctica in winter. Relying on relatively cost effective, fibered telecom lasers, the costs associated with all the components for the instrument are estimated to be slightly higher than a commercial Picarro analyser, in particular due to the implementation of two cavities. While developing a field version will require some additional engineering resources, dedicated instruments are needed to respond to the very niche requirements for Polar Regions.

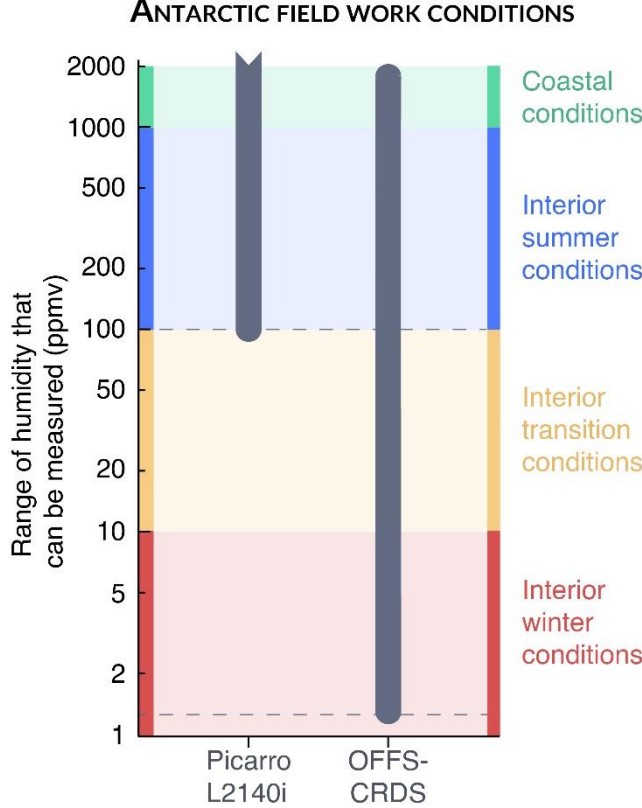

***Figure 9:*** *Range of humidities at which the instrument can measure isotopic composition with a precision better than 1 ‰.*

One limitation to determine the true precision and accuracy of the VCOF-CRDS instrument comes from the vapour generation from both the dedicated instrument (Leroy-Dos Santos et al., 2021) or from the outgassing of the tubes. Indeed, at such low humidity, it is challenging to limit the variations of the humidity levels below 1 ppmv, which can account for a significant amount of the total humidity levels below 100 ppmv. The use of several instruments monitoring the same water vapour would be needed to disentangle the various contributions to the variability coming from the water generation from the one coming from the measurement.


The current instrument cannot be deployed to the field due to its size and weight (1 m$^3$, 230 kg), the fragility of the VCOF source (the glass cavity is suspended by fragile ceramic rods, vacuum must be maintained), and the bulkiness and the frailty of the control electronics (instruments must be extremely isolated for static shocks due to the lack of electric ground with the thick ice layer). The performances of the instrument in the field should be roughly the same, provided that the temperature

stabilisation of the instrument is as good as in the AC rooms of the lab. Indeed, the proof of concept for the frequency auto-referencing shows that the instrument will not suffer drift associated with change of the laser source wavelength. Another caveat before the instrument can be deployed for field work is to take into account interferences from other species. For instance, taking into account the impact of methane absorption features at 7199.95 cm$^{-1}$ and 7200.03 cm$^{-1}$ will require adding an extra spectral point for Antarctic field study as the methane absorption should be as strong as the water ones at humidity

level around 1 ppmv, following a similar approach than (Chaillot et al., 2023).

The determination of physical parameters associated with the different heavy isotopes is also limited by instrumental capabilities. Currently, several studies attempted to measure the fractionation associated with gaseous solid phase transitions and ended with contradictory results at temperature below -30 °C (Ellehoj et al., 2013; Lamb et al., 2017; Majoube, 1971).

Better determinations of these fractionation coefficients are key for isotope enabled climate models (Risi et al., 2008; Werner et al., 2011) which use these parametrisation throughout the water cycle, with temperature often much lower than -30 °C, especially in high altitude cloud microphysics processes. Using dedicated laboratory experiments, the VCOF-CRDS instrument would be well suited to determine the equilibrium fractionation coefficient down to -80°C, shedding light on fractionation processes at temperature range relevant for cloud microphysics or in Polar Regions.

**4. Conclusion**

We build up a infrared spectrometer based on relatively cheap telecom components enhanced by high performances of optical feedback frequency stabilisation and cavity ring down spectroscopy. This instrument demonstrates a precision and a stability of 0.01 ‰ for δ$^{18}$O and 0.1 ‰ for d-excess at extremely low humidities such as found in central Antarctica for durations longer than 2 days. A key element to ensure limited drift even outside of the confine of a fully equipped spectroscopy lab is the use

of the Lamb-dip feature as a frequency reference. This shows that the instrument is able to reach the same level of precision without any external validation of the frequency of the laser source. While this instrument cannot be transported to Antarctica, by supporting measurement down to 1 ppmv of humidity, this technique shows great potential to study the isotopic exchanges all year long in Antarctica where temperature can reach -80 °C during the polar night.

 **Appendix A: Evaluation of the stability of the generated vapour levels**

The humidity generator used here is based on the instrument described in (Leroy-Dos Santos et al., 2021). The humidity levels it generates, while extremely stable, are limited to a stability of roughly 20 ppmv over one hour. Due to the high precision of the infrared spectrometer discussed in this manuscript, it is possible that the drift observed on the isotopic composition here is linked with instability in the generated water humidity, and thus, isotopic composition associated with limited variations in the

water and air inflow in the humidity generator. Figure A1a) shows the variations of humidity over the 8-day period that was used to produce the Allan standard deviation curve at 400 ppmv in Figure 4a. Over the course of 8 days, the measured humidity only varied by 10 ppmv, principally during refill of the syringe of the humidity generator (Leroy-Dos Santos et al., 2021). In addition, a drift of roughly 2.5 ppmv over the 8 days is visible on the time series. This drift is well visible on the Allan standard deviation plots (Fig. A1b). The very similar shape of the increase of the Allan standard deviation of specific humidity around

1 hour is very similar to the bump in the $\delta^{18}O$ Allan standard deviation (Fig. 4a) which suggests that the instability in the generated vapour sample could have created excess noise which is not linked with the spectrometer.

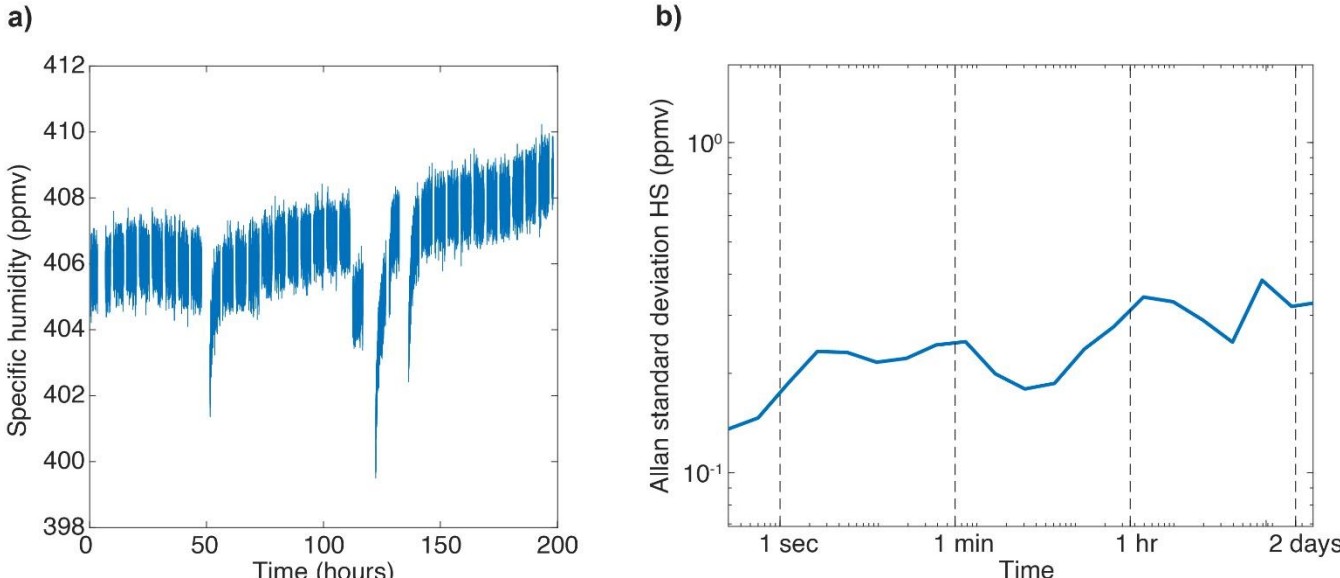

***Figure A1:*** *a)Specific humidity time series during an 8-day monitoring of a stable isotopic sample, b) Allan standard deviation*
*plots of the specific humidity measurements*

## Appendix B: Comparison of the Allan standard deviation of $\delta^{18}O$ and $\delta D$

Figure 4 and the result section focuses on $\delta^{18}O$ and d-excess because of the excess noise found in the $\delta^{18}O$ measurements due to the misalignment of the measurement point off the centre of the $H_2^{18}O$ transition. In fact, we expect the drift of d-excess to be dominated by the drift of $\delta^{18}O$ due to the excess noise. Here, we show that this is relatively true at short time scales (below one hour, Fig. B1). At longer time scales, the excess noise from $\delta D$ leads to a flat response of the d-excess Allan standard deviation.

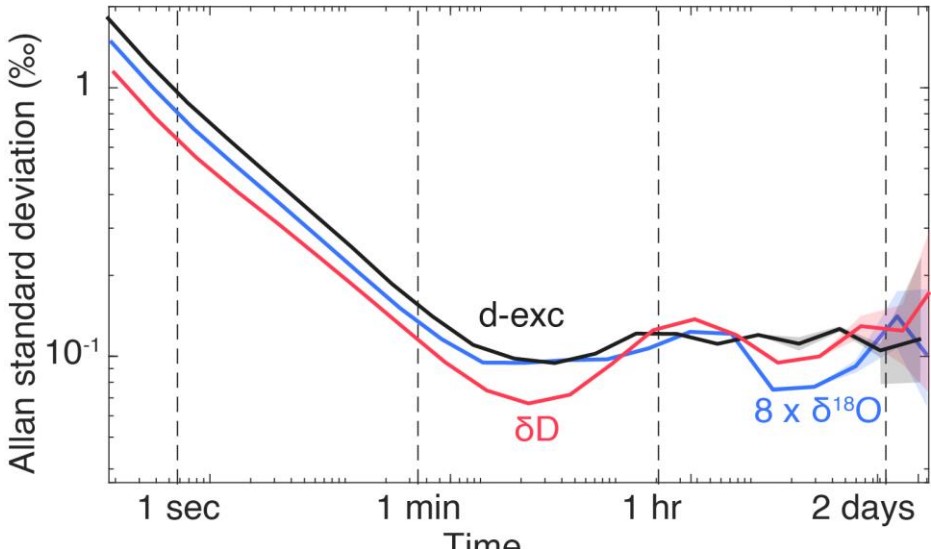

*Figure B1: Comparison of the Allan standard deviations of $\delta^{18}O$ (scaled by a factor 8 to ease the comparison), $\delta D$, and d-excess realised at 400 ppmv.*

***Author contributions.*** *AL and SK organised the project, MC, TS, JC and SK built the instrument, FP and BB support the project and the measurements, MC wrote the manuscript with help of all the co-authors.*

***Acknowledgments.*** *The research leading to these results has received funding from the European Research Council Combiniso project (FP7/2007- 2013)/RC grant agreement number 306045 and SAMIR project (HORIZON)/RC grant agreement number 101116660. We thank Erik Kerstel, Daniele Romanini, Johannes Burkart, and Peter Cermak for our fruitful discussions which helped improve the manuscript. We would like to thank the editor, Pierre Herckes, and two anonymous reviewers and Béla Tuzson for their comments which improved the quality of the manuscript.*

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
