# Peer review of "Reliable water vapour isotopic composition measurements at low humidity using frequency stabilised cavity ring down spectroscopy"

_EGUsphere, 2023_

## Referee Comment (RC1)

A Review of: "New infrared spectroscopy instrument for reliable low humidity water vapour isotopic composition measurements" (2023-2457), submitted by Mathieu Casado et al. to Atmospheric Measurement Techniques (via EGUsphere)

This manuscript puts forward a new measurement setup for measuring stable water isotopes in extremely dry air. Though the core of the measurement technique is based off existing technology, it combines it with additional technology in a novel way. Additionally, the ever-present issue to combat instrument drift in CRDS instruments is addressed in an innovative and creative manner, via frequency referencing with a physical constant.

In this review, I will focus mainly on the use of CRDS technology to measure stable water isotopes in low humidity environments as this is closest to my area of expertise. However, some of the finer details of the spectroscopy are beyond my expertise, specifically Sect 2.3. Nevertheless, I will do my best to fairly assess all sections of the manuscript.

Overall, I very much enjoyed reading and learning about this new development in stable water isotope measurement. The science presented certainly warrants publication, and the manuscript is nearly there. With only a few revisions and better management of the readers' expectations, this work has the potential to become a foundational piece of literature in the further advancement of CRDS technology and the stable water isotope research community.

**Major comments**

1. My most pressing critique concerns the assertion that the instrument accurately measures isotopic signal down to 1 ppmv. The abstract and introduction mention this capability multiple times, which would be an outstanding achievement and got me very excited to read more of the manuscript. However, in Sect. 3 (Results), the authors show that the lowest experimental measurements are at 25 ppmv and that the precision at 1 ppmv was extrapolated. This undercuts the very good work by the authors to make accurate measurements at 25 ppmv, which is a formidable achievement in itself. Therefore, this claim needs to be rephrased here and elsewhere to comply with the actual measurement results that are shown. For example, I would suggest modifying the initial claims of "1 ppmv" to "10 ppmv". I believe that this order of magnitude better reflects the experiments conducted by the authors with the calibration unit available to them. I also suggest moving the sentences discussing extrapolation down to 1 ppmv to Sect. 4 (Discussion), with the context of vapour generation limitations. To be clear, I don't disbelieve the authors that the instrument could be capable of accurate measurements at 1 ppmv if given the chance, however if it remains a "*could*" it should be presented as such.

This should also be reflected in Fig. 8, with some differentiation needed in the bottom section of the OFFS-CRDS grey bar, below 10 ppmv (perhaps a lighter shade of grey or dashed outlining). With this measurement expectation established early in the manuscript, I believe readers will still find the work exciting, while knowing what to anticipate from the instrument.

2. I think that the authors should re-consider how they establish the comparison between the new instrument and the commercial CRDS analyzer. As it stands now, the manuscript portrays the new instrument as an alternative to a commercial device, using a Picarro L2130i as a benchmark. As this is an instrument that many in the field would be familiar with, it serves as a relatable reference for readers. However, unlike the Picarro, the new instrument cannot be deployed to the field in its current form and is much larger and more sensitive in regard to handling than the simple benchtop analyzers. This does not detract from the quality of the scientific advance put forward by the authors but establishing the reality of the instrument in Sect. 1 (Introduction) with a short remark (big, bulky, and fragile) would better manage the expectations of readers.

**Minor comments**

**1. Introduction**

The authors do a satisfactory job of conveying the utility of stable water isotopes and the gaps present with their observations. Additionally, an overview of the technology used to detect stable water isotopes is well covered. Numerous, relevant references are made to past bodies of work. Aside from the two issues mentioned in the Major comments, the authors do well to situate readers for the work that follows.

**2. Methods**

2.1 The authors detail the 3 main components and their workings in a logical manner, paralleling the structure of Fig. 1. While I compliment the authors on the appealing design of the figure, I'm left wondering why the authors didn't try to connect it more to the text, as they only generally refer to it once (L66). I'd suggest re-considering the level of detail included in Fig. 1, making it even more of a bare schematic for the main text. Then the more detailed version could be included in the Appendix.

2.2 Fig. 2 is very nicely designed, with a helpful connection between subfigures a) and b).

2.3 Fig. 3 suffers from the specific placement of the "Lamb-dip" label. At first, I thought I had misunderstood what the Lamb-dip was, and that the sharp decrease and subsequent

recovery in the red line at 90-140 hours was somehow evidence of the Lamb-dip. I believe this misunderstanding should be fixed by changing the location of the label to around the (40,-0.05) area.

**3. Results**

3.1 The authors separate the findings of their experiments into precision and accuracy, with an additional highlight on the frequency auto-referencing. Aside from the extrapolation item mentioned in my Major comments, the methodology for determining instrument precision and stability is well documented.

3.2 Though it also naturally fits in the explanation of the experimental results, the influence of the vapour generator (L172) and the difficulty of disentangling its impact should be mentioned in Sect. 4 (Discussion), including the lower limits of vapour generation.

3.3 Fig. 4 is also well-designed, with creative linkages drawn between the subpanels, which help to orient the reader. However, the coloured surface in b) is a bit distracting for this purpose. Perhaps a greyscale colourmap would better show the lines drawn for the connections to subfigures a) and c).

3.4 It may also be worth mentioning that the exact isotope-humidity response is unique to each individual analyzer (Weng et al., 2020), so an instrument without any such a response is even more valuable.

3.5 Though complex, Fig. 7 nicely visualizes the concepts presented in the text, especially b), d), and f). I would only suggest keeping the labelling consistent with Fig. 3, since the red "Self referenced" is the Lamb dip method and the blue "External reference" is the optical comb method (right?).

**4. Discussion**

4.1 The authors do well to put forward how the current work would fit into previous work. The speculated benefit that the instrument would bring to Antarctic field research is very exciting.

4.2 I appreciate that the authors also connect the potential benefit of these low humidity measurements to a concrete scientific aim, which is the fractionation between gas and solid phase at such extremely low temperatures. If it is actually feasible, I suggest adding a brief comment on the possibility that the new instrument could measure this fractionation in lab experiments, inside which it seems the instrument would be most comfortable in its current iteration. Such an application would be a further demonstration of its value and utility.

4.3 As the instrument is being compared to a commercially available Picarro, consider a qualitative comparison of the financial cost, even if this comparison is as simple as "much more/less" or similar.

**5. Conclusion**

5.1 The authors concisely present the key findings of the work, though the term "cheap telecom" or similar was absent from the rest of the manuscript. As mentioned above, including some indication of cost in Sect. 4 (Discussion) would support this conclusion.

5.2 The viability of the frequency reference technique is deservedly highlighted in its own right. By finishing with a statement focusing on the lower measurement limit of the instrument, the authors nicely echo the title of the work, and end on a definite.

**6. Appendices & References**

6.1 As mentioned above, the influence that the vapour generator might have on the experiment should be touched upon in Sect, 4 (Discussion), but the authors made good use of Appendix A to contain the more specific details.

6.2 A similar point can be made for Appendix B, which concisely justifies why the authors focused on $\delta^{18}O$ and not $\delta D$. However, I would suggest renaming the current appendices to B and C, and making a new Appendix A, which contains the detailed breakdown of the current Fig. 1.

**Detailed comments**

- L21: Consider "stable water isotopes" as this word order is used in L159.
- L39: This sentence is rather long, and the knowledge gap (no instrument able) is buried in the middle. Consider separating it into two. For example: "Ritter et al., 2016). This is despite attempts to develop a new generation"
- L41: Consistent spacing between value and unit "20 ppmv". Not necessary for % or ‰
- L54: 0.06 Pa
- L55: -90 °C
- L56: "or on other planets."
- L71: Include abbreviation expansion in the figure caption (e.g. MZM).
- L81: "which enables us to tune" or "which enables tuning of the frequency"
- L85: 0.9 mW
- L94: 0.01 mbar

- L100: Consider explicitly connecting the mode established in the previous paragraph end sentence to the opening of the new paragraph. For example, "In full spectrum mode, the instrument has a high spectral resolution and can be used"
- L101: Does the word "Here" refer to the high spectral resolution mode mentioned in the previous sentence, or is it being used to establish the other high pace mode used in this work? Again, consider using explicit reference to the previously established mode names introduced at the start of Sect. 2.2 (full spectrum vs. high pace)
- L101: The phrase "water vapour isotopic," doesn't really fit here. Maybe "isotopic water vapour," or "water vapour isotopes," fits better.
- L104: Remove the word of: "This is done by "jumping" exactly one FSR"
- L104: Expand FSR abbreviation
- L105: Consider re-phrasing to "the spectral resolution is only multiples of the FSR"
- L108: "which leads to additional"
- L125: I would personally recommend minimizing the use of latin phrases unless absolutely necessary or further explained in the text. Consider instead "tackled by an empirically established drift correction" or similar
- L126: "frequency comb which is itself locked on the"
- L130: "by very small linewidth"
- L140: "and then measure the Lamb dip"
- L143: Fig. 3 would be even better understood if there would be clarification that the figure is the result of a 6 day experiment somewhere near this sentence. For example, "Across a 140 hour experiment, we measure the frequency deviation obtained with the Lamb dip method…"
- L153: "in Leroy-Dos Santos et al. (2021)."
- L161: Is there an upper limit to the humidity that the instrument can measure? In other words, is the upper measurement limit comparable to the Picarro, or does the special configuration of the instrument impose a limitation? A brief sentence in Sect. 4 (Discussion) would do.
- L170: Appendix B
- L185: "from Fig. 3 of Leroy-Dos Santos et al. (2021),"
- L186: "from Casado et al. (2016),"
- L190: "While we were not able"
- L200: "response until humidities around 200 ppmv". Consider instead "flat humidity response for humidities above 200 ppmv."
- L209: Remove "as detailed by". Otherwise, "as detailed by Casado et al. (2016) and…"
- L210: 500 ppmv
- L210: 100 ppmv

- L214: 100 ppmv
- L220: "in Leroy-Dos Santos et al. (2021)
- L232: 50 MHz
- L232: 1 MHz
- L232: 10 MHz
- L243: "measured Lamb dip features every hour."
- L257: "Lamb dip measurements"
- L263: "would mitigate a large part"
- L273: -40 °C
- L276: Consider "In inland Antarctica, some campaigns monitored isotopic water vapour composition but were"
- L280: "use of the high sensitivity"
- L280: "to an Antarctic field station"
- L283: As mentioned in my opening comments, I think "10 times lower" is more accurate. But this would be a good point to speculate on the potential precision down to 1 ppmv.
- L284: -80 °C
- L286: Remove one of the "monitoring". Consider "all the hurdles that limit water vapour isotopic composition monitoring in the coldest…" or "all the hurdles that limit the monitoring of water vapour isotopic composition in the coldest…"
- L293: 230 kg
- L301: Remove either "Indeed" or "currently"
- L301: "gaseous solid phase"
- L302: -30 °C
- L305: -30 °C
- L307: I'm not sure what is meant by "hyped". Would maybe "enhanced" or "complimented" fit better?
- L309: durations
- L314: -80 °C
- L318: 1 hour
- L330: Specific humidity
- L387: spectroscopy of $H_2S$
- L445: "Available from: %3CGo". What is %3CGo?
- L452: "$\delta^{18}O$ and $\delta D$"

---

## Author Comment (AC1)

Dear editor,

We would like to thank you and the reviewer 3 for their comments. We have been working toward a new version of the manuscript taking their respective comments into account. We include the comments from the reviewers in black, our responses in blue, and the modifications to the manuscript in red in this response file.

All the best,

On the behalf of all the co-authors,

Mathieu Casado

**Reviewer #1:**

A Review of: "New infrared spectroscopy instrument for reliable low humidity water vapour isotopic composition measurements" (2023-2457), submitted by Mathieu Casado et al.to Atmospheric Measurement Techniques (via EGUsphere) This manuscript puts forward a new measurement setup for measuring stable water isotopes in extremely dry air. Though the core of the measurement technique is based off existing technology, it combines it with additional technology in a novel way. Additionally, the ever-present issue to combat instrument drift in CRDS instruments is addressed in an innovative and creative manner, via frequency referencing with a physical constant.

In this review, I will focus mainly on the use of CRDS technology to measure stable water isotopes in low humidity environments as this is closest to my area of expertise. However, some of the finer details of the spectroscopy are beyond my expertise, specifically Sect 2.3. Nevertheless, I will do my best to fairly assess all sections of the manuscript. Overall, I very much enjoyed reading and learning about this new development in stable water isotope measurement. The science presented certainly warrants publication, and the manuscript is nearly there. With only a few revisions and better management of the readers' expectations, this work has the potential to become a foundational piece of literature in the further advancement of CRDS technology and the stable water isotope research community.

We thank the reviewer 1 for evaluating carefully our manuscript and will address their comments accordingly in order to improve the manuscript.

Major comments

1. My most pressing critique concerns the assertion that the instrument accurately measures isotopic signal down to 1 ppmv. The abstract and introduction mention this capability multiple times, which would be an outstanding achievement and got me very excited to read more of the manuscript. However, in Sect. 3 (Results), the authors show that the lowest experimental measurements are at 25 ppmv and that the precision at 1 ppmv was extrapolated. This undercuts the very good work by the authors to make accurate measurements at 25 ppmv, which is a formidable achievement in itself.

Therefore, this claim needs to be rephrased here and elsewhere to comply with the actual measurement results that are shown. For example, I would suggest modifying the initial claims of "1 ppmv" to "10 ppmv". I believe that this order of magnitude better reflects the experiments conducted by the authors with the calibration unit available to them. I also suggest moving the sentences discussing extrapolation down to 1 ppmv to Sect. 4 (Discussion), with the context of vapour generation limitations. To be clear, I don't disbelieve the authors that the instrument could be capable of accurate measurements at 1 ppmv if given the chance, however if it remains a "could" it should be presented as such. This should also be reflected in Fig. 8, with some differentiation needed in the bottom section of the OFFS-CRDS grey bar, below 10 ppmv (perhaps a lighter shade of grey or dashed outlining). With this measurement expectation established early in the manuscript, I believe readers will still find the work exciting, while knowing what to anticipate from the instrument.

This is a completely fair comment (which was also noted by the two other reviewers). Since the manuscript was submitted, the instrument has been transferred to another institute. As we realised tests to validate that the performances of the instruments were unchanged, we developed a new set up to produce Allan variance at humidity levels below 25 ppmv. Indeed, the first evaluation was solely made using the water vapour generator described in (Leroy-Dos Santos et al., 2021) which was only tested down to 70 ppmv in the original manuscript. We showed that the instrument performed well down to 20 ppmv, but could not explore much lower humidity levels due to outgassing inside the water vapour generator.

Now, we included direct measurement from the dry air bottles (Air Products, 3 ppm) to reach lower humidity amounts. In order to limit outgassing to a minimum, we included the smallest tubings between the instrument and the bottle and measured the water traces which outgassed from the tube between the bottle and the instruments. We were able to produce stable water levels at humidity levels ranging from 3.8 to 10 ppmv to confirm the performances of the instrument at low humidity. The results were included in Section 3.1 (lines 246 to 255):

> "While we were not able to generate stable moisture flux at 1 ppmv using the humidity generated described in (Leroy-Dos Santos et al., 2021), we extrapolated the precision expected after two minutes at humidity lower than 25 ppmv by fitting the data with a power law and find a precision of 1.5 ‰ at roughly 1 ppmv. We also directly connected the instrument to dry air bottle controlling the humidity from the outgassing of the tubes connecting the bottle to the instrument (see Methods, Section 2.4) and evaluated the precision at 3.8, 4.2, 6.5, and 11 ppmv. Since this method to generate water standard is extremely dependent on temperature variations, it is only useful as an upper bound of the precision of the instrument as the humidity levels are relatively variable (standard deviation of the humidity larger than 10% of the humidity content), and thus, these datapoints where not included in the power law fit. We find precisions ranging from 0.5 to 0.7 ‰ after two minutes (Fig. 4c) which agree relatively well with the power law defined at larger humidity levels."

We believe that with the additional datapoints down to 3.8ppmv, the use of an extrapolation of the precision down to 1 ppmv is justified, but we have been careful to include that the capability to measure down to 1 ppmv was only extrapolated throughout the manuscript.

We also added a new section 2.4 to detail the two different ways that were used to generate the water vapour standards used in the manuscript (lines 182 to 198):

> "2.4 Generation of water vapour standards
>
> We used the water vapour generator instrument described in (Leroy-Dos Santos et al., 2021) to generate stable water levels to evaluate the response of the new infrared spectrometer to varying humidity levels. We used dry air bottles with less than 3 ppmv of water to supply the gas to the instrument. We used laboratory standards of water with isotopic composition varying from 0 to -60 ‰ to supply with water of known isotopic composition. The water vapour generator shows relatively good performances down to humidity around 20 ppmv where outgassing from the instrument itself limits its performance. As a result, we used the water vapour generator to evaluate the precision as a function of humidity (section 3.1) and the humidity response of the instrument (section 3.2) down to 25 ppmv.

To obtain an evaluation of the precision of the instrument below these humidity levels, we connected a dry air bottle directly to the instrument and regulated the produced humidity levels from outgassing of the tube by changing the flux of dry air to an exhaust connected to a sonic nozzle. Using this method, humidity levels down to 4 ppmv could be reached. In this case, the isotopic composition is not known so it cannot be used to evaluate the accuracy of the instrument, only how stable the instrument is capable to measure relatively stable isotopic composition of outgassing water from the tube walls. As the temperature of the tube and the dry air canister were not regulated nor monitored, the resulting water vapour was relatively variable (variations around 10% of the produced humidity level), which lead to potential variability in the isotopic composition, leading itself to relatively high Allan standard deviations not necessarily linked with poor performance of the instrument."

We also changed the abstract (lines 19 to 21):

"This set up yields an isotopic composition precision below 1‰ at water mixing ratios down to 4 ppmv, which suggest an extrapolated precision in δ18O of 1.5‰ at 1 ppmv in two minutes."

and the introduction to reflect this update in the manuscript:

"conditions, and should reach satisfactory performances at humidity around 1 ppmv"

2. I think that the authors should re-consider how they establish the comparison between the new instrument and the commercial CRDS analyzer. As it stands now, the manuscript portrays the new instrument as an alternative to a commercial device, using a Picarro L2130i as a benchmark. As this is an instrument that many in the field would be familiar with, it serves as a relatable reference for readers. However, unlike the Picarro, the new instrument cannot be deployed to the field in its current form and is much larger and more sensitive in regard to handling than the simple benchtop analyzers. This does not detract from the quality of the scientific advance put forward by the authors but establishing the reality of the instrument in Sect. 1 (Introduction) with a short remark (big, bulky, and fragile) would better manage the expectations of readers.

This is a very good point. We now conclude the introduction with this sentence (lines 69 to 71):

"While the current prototype cannot be deployed to the field due to its relative bulkiness (1m3), weight, and fragility, this manuscript intends as providing a proof of concept for the application of this technology for laboratory and field monitoring of water vapour isotopic composition at humidity levels down to 1 ppmv."

Minor comments

1. Introduction

The authors do a satisfactory job of conveying the utility of stable water isotopes and the gaps present with their observations. Additionally, an overview of the technology used to detect stable water isotopes is well covered. Numerous, relevant references are made to past bodies of work. Aside from the two issues mentioned in the Major comments, the authors do well to situate readers for the work that follows.

Changes to the introduction and the abstract has been included to reflect the major comments of the reviewer.

2. Methods

2.1 The authors detail the 3 main components and their workings in a logical manner, paralleling the structure of Fig. 1. While I compliment the authors on the appealing design of the figure, I'm left wondering why the authors didn't try to connect it more to the text, as they only generally refer to it once (L66). I'd suggest re-considering the level of detail included in Fig. 1, making it even more of a bare schematic for the main text. Then the more detailed version could be included in the Appendix.

Additional references to the different subfigures of Figure 1 have been included in the text. We prefer maintaining this version of the figure in the manuscript as we believe it is useful to have a detailed description of the important *logic* functions of the instrument.

2.2 Fig. 2 is very nicely designed, with a helpful connection between subfigures a) and b).

Thank you.

2.3 Fig. 3 suffers from the specific placement of the "Lamb-dip" label. At first, I thought I had misunderstood what the Lamb-dip was, and that the sharp decrease and subsequent recovery in the red line at 90-140 hours was somehow evidence of the Lamb-dip. I believe this misunderstanding should be fixed by changing the location of the label to around the (40,-0.05) area.

Taken into account.

3. Results

3.1 The authors separate the findings of their experiments into precision and accuracy, with an additional highlight on the frequency auto-referencing. Aside from the extrapolation item mentioned in my Major comments, the methodology for determining instrument precision and stability is well documented.

Thank you.

3.2 Though it also naturally fits in the explanation of the experimental results, the influence of the vapour generator (L172) and the difficulty of disentangling its impact should be mentioned in Sect. 4 (Discussion), including the lower limits of vapour generation.

Discussion about the vapour generation issues has been added in Section 4 (lines 391 to 395): "One limitation to determine the true precision and accuracy of the new VCOF-CRDS instrument comes from the vapour generation from both the dedicated instrument (Leroy-Dos Santos et al., 2021) or from the outgassing of the tubes. Indeed, at such low humidity, it is challenging to limit the variations of the humidity levels below 1 ppmv, which can account for a significant amount of the total humidity levels below 100 ppmv. The use of several instruments monitoring the same water vapour would be needed to disentangle the variability coming from the water generation from the one coming from the measurement."

3.3 Fig. 4 is also well-designed, with creative linkages drawn between the subpanels, which help to orient the reader. However, the coloured surface in b) is a bit distracting for this purpose.

Perhaps a greyscale colourmap would better show the lines drawn for the connections to subfigures a) and c).

We acknowledge that the colourmap is quite *flashy* compared to the rest of the figure. We tried to reduce the opacity of the colourmap so it is less distracting as a common ground between greyscale and the flashy colours.

3.4 It may also be worth mentioning that the exact isotope-humidity response is unique to each individual analyzer (Weng et al., 2020), so an instrument without any such a response is even more valuable.

We agree and included a sentence to discuss this aspect. We also mention that if the isotope-humidity response is unique to each analyser, this could also be true for our instrument, so in all fairness, a validation of these results with another VCOF-CRDS analyser would be needed to confirm that this low isotope humidity response was not luck. Additional text goes as follow (lines 289 to 292):

> "Additionally, for Picarro instruments, the humidity response is very variable in between individual instruments (Weng et al., 2020), and needs to be evaluated for each new analyser, as well as each time an analyser is deployed (Leroy-Dos Santos et al., 2021). The limited humidity response of our new setup is therefore extremely valuable but would need to be validated on a second VCOF-CRDS instrument to be confirmed"

3.5 Though complex, Fig. 7 nicely visualizes the concepts presented in the text, especially b), d), and f). I would only suggest keeping the labelling consistent with Fig. 3, since the red "Self referenced" is the Lamb dip method and the blue "External reference" is the optical comb method (right?).

As Figure 7g) was not extremely busy, we merge both labelling, explaining that the "Self referenced" was using the lamb dip and that the "External reference" the optical frequency comb.

4. Discussion

4.1 The authors do well to put forward how the current work would fit into previous work. The speculated benefit that the instrument would bring to Antarctic field research is very exciting.

Thank you.

4.2 I appreciate that the authors also connect the potential benefit of these low humidity measurements to a concrete scientific aim, which is the fractionation between gas and solid phase at such extremely low temperatures. If it is actually feasible, I suggest adding a brief comment on the possibility that the new instrument could measure this fractionation in lab experiments, inside which it seems the instrument would be most comfortable in its current iteration. Such an application would be a further demonstration of its value and utility.

This is actually the topic of a PhD we are supervising now and hope to have results soon on the topic. In the meantime, we included a sentence in this direction in the discussion (lines 413 to 415):

> "Using dedicated laboratory experiments, the new VCOF-CRDS instrument would be well suited to determine the equilibrium fractionation coefficient down to -80°C,

shedding light on fractionation processes at temperature range relevant for cloud microphysics or in Polar Regions."

4.3 As the instrument is being compared to a commercially available Picarro, consider a qualitative comparison of the financial cost, even if this comparison is as simple as "much more/less" or similar.

We agree, and suggest including the following sentence to discuss these aspects more in depth in the discussion (lines 382 to 386):

"Relying on relatively cheap, fibered lasers which are commonly build for telecommunication, the costs associated with all the components for the new instrument are estimated to be slightly higher than a commercial Picarro analyser, in particular due to the implementation of two cavities. While developing a field version will require some additional engineering resources, dedicated instruments are needed to respond to the very niche requirements for Polar Regions."

5. Conclusion

5.1 The authors concisely present the key findings of the work, though the term "cheap telecom" or similar was absent from the rest of the manuscript. As mentioned above, including some indication of cost in Sect. 4 (Discussion) would support this conclusion.

This is true, see previous comments for additions to the discussion that would support this statement.

5.2 The viability of the frequency reference technique is deservedly highlighted in its own right. By finishing with a statement focusing on the lower measurement limit of the instrument, the authors nicely echo the title of the work, and end on a definite.

Thank you.

6. Appendices & References

6.1 As mentioned above, the influence that the vapour generator might have on the experiment should be touched upon in Sect, 4 (Discussion), but the authors made good use of Appendix A to contain the more specific details.

See the dedicated comment for the addition about the vapour generator in the discussion.

6.2 A similar point can be made for Appendix B, which concisely justifies why the authors focused on $\delta 18O$ and not $\delta D$. However, I would suggest renaming the current appendices to B and C, and making a new Appendix A, which contains the detailed breakdown of the current Fig. 1.

As we referred more to the Figure 1 in the updated version of the manuscript, we've for now taken the option of not moving it in the appendix.

Detailed comments

• L21: Consider "stable water isotopes" as this word order is used in L159.

Taken into account.

• L39: This sentence is rather long, and the knowledge gap (no instrument able) is buried in the middle. Consider separating it into two. For example: "Ritter et al., 2016). This is despite attempts to develop a new generation"

Taken into account.

• L41: Consistent spacing between value and unit "20 ppmv". Not necessary for % or ‰

We've tried to systematically include a spacing before "ppmv", and units in general, following guidelines from Copernicus

• L54: 0.06 Pa

Taken into account.

• L55: -90 °C

Taken into account.

• L56: "or on other planets."

Taken into account.

• L71: Include abbreviation expansion in the figure caption (e.g. MZM).

Taken into account.

• L81: "which enables us to tune" or "which enables tuning of the frequency"

Taken into account.

• L85: 0.9 mW

Taken into account.

• L94: 0.01 mbar

Taken into account.

• L100: Consider explicitly connecting the mode established in the previous paragraph end sentence to the opening of the new paragraph. For example, "In full spectrum mode, the instrument has a high spectral resolution and can be used"

Taken into account.

• L101: Does the word "Here" refer to the high spectral resolution mode mentioned in the previous sentence, or is it being used to establish the other high pace mode used in this work? Again, consider using explicit reference to the previously established mode names introduced at the start of Sect. 2.2 (full spectrum vs. high pace)

This makes sense, and has been implemented in the revised version of the manuscript.

• L101: The phrase "water vapour isotopic," doesn't really fit here. Maybe "isotopic water vapour," or "water vapour isotopes," fits better.

The sentence was modified to: "to monitor water vapour isotopic composition"

• L104: Remove the word of: "This is done by "jumping" exactly one FSR"

Taken into account.

• L104: Expand FSR abbreviation

Taken into account.

• L105: Consider re-phrasing to "the spectral resolution is only multiples of the FSR"

Taken into account.

• L108: "which leads to additional"

Taken into account.

• L125: I would personally recommend minimizing the use of latin phrases unless absolutely necessary or further explained in the text. Consider instead "tackled by an empirically established drift correction" or similar

We understand, and really wanted to insist that these corrections are usually done after the measurement took place. We suggest the following latin-free updated sentence (lines 144-145):

> "is usually tackled by post-correction of empirically established drift of the measurement of the isotopic composition itself".

• L126: "frequency comb which is itself locked on the"

Taken into account.

• L130: "by very small linewidth"

Taken into account.

• L140: "and then measure the Lamb dip"

Taken into account.

• L143: Fig. 3 would be even better understood if there would be clarification that the figure is the result of a 6 day experiment somewhere near this sentence. For example, "Across a 140 hour experiment, we measure the frequency deviation obtained with the Lamb dip method..."

Taken into account.

• L153: "in Leroy-Dos Santos et al. (2021)."

Taken into account.

• L161: Is there an upper limit to the humidity that the instrument can measure? In otherwords, is the upper measurement limit comparable to the Picarro, or does the special configuration of the instrument impose a limitation? A brief sentence in Sect. 4 (Discussion) would do.

To be tested

• L170: Appendix B

Taken into account.

• L185: "from Fig. 3 of Leroy-Dos Santos et al. (2021),"

Taken into account.

• L186: "from Casado et al. (2016),"

Taken into account.

• L190: "While we were not able"

Taken into account.

• L200: "response until humidities around 200 ppmv". Consider instead "flat humidity response for humidities above 200 ppmv."

Taken into account.

• L209: Remove "as detailed by". Otherwise, "as detailed by Casado et al. (2016) and..."

Taken into account.

• L210: 500 ppmv

• L210: 100 ppmv

• L214: 100 ppmv

All three taken into account.

• L220: "in Leroy-Dos Santos et al. (2021)

Taken into account.

• L232: 50 MHz

• L232: 1 MHz

• L232: 10 MHz

All three taken into account.

• L243: "measured Lamb dip features every hour."

Taken into account.

• L257: "Lamb dip measurements"

Taken into account.

• L263: "would mitigate a large part"

Taken into account.

• L273: -40 °C

Taken into account.

• L276: Consider "In inland Antarctica, some campaigns monitored isotopic water vapour composition but were"

Taken into account.

• L280: "use of the high sensitivity"

Taken into account.

• L280: "to an Antarctic field station"

Taken into account.

• L283: As mentioned in my opening comments, I think "10 times lower" is more accurate. But this would be a good point to speculate on the potential precision down to 1 ppmv.

Considering the new experiments showing the ability to measure effectively with a precision of roughly 0.5‰ at 4 ppmv, and the validation of the extrapolation below 10 ppmv of humidity, we believe that we can maintain the ability to measure at 1 ppmv. Note that to be more coherent with the Picarro instrument, we did an averaging to 2 minutes, which lead to the precision now being 1.5‰ at 1 ppmv, which still seems acceptable.

• L284: -80 °C

Taken into account.

• L286: Remove one of the "monitoring". Consider "all the hurdles that limit water vapour isotopic composition monitoring in the coldest..." or "all the hurdles that limit the monitoring of water vapour isotopic composition in the coldest..."

Taken into account.

• L293: 230 kg

Taken into account.

• L301: Remove either "Indeed" or "currently"

Taken into account.

• L301: "gaseous solid phase"

Taken into account.

• L302: -30 °C

• L305: -30 °C

Both taken into account.

• L307: I'm not sure what is meant by "hyped". Would maybe "enhanced" or "complimented" fit better?

Taken into account.

• L309: durations

Taken into account.

• L314: -80 °C

Taken into account.

• L318: 1 hour

Taken into account.

• L330: Specific humidity

Taken into account.

• L387: spectroscopy of H2S

Taken into account.

• L445: "Available from: %3CGo". What is %3CGo?

Taken into account.

• L452: "δ18O and δD"

This is for the latex bibliography, unfortunately, I cannot change this at this point to avoid crashing all my latex projects.

**Reviewer #2:**

The manuscript of Casado *et al*. addresses the urging and relevant topic of high-precision isotope ratio measurements of atmospheric water vapour with special target on extremely low amount fractions (~1 ppm). The authors present the development of a highly complex spectroscopic setup based on a high-finesse cavity ring down (CRDS) technique in combination with frequency stabilized laser source using the optical feedback from a V-shaped cavity. This approach allows for outstanding precision and long-term stability, compared to the performance of commercial instruments, in particular to Picarro L2140i. Thus, the proposed analytical tool can be highly relevant e.g. for the polar atmospheric research.

Despite the very impressive instrumental developments, the manuscript has a few caveats in presenting this otherwise very promising topic (see general comments for details). The authors should consider consolidating the wording, language, and the structure of their manuscript. A more precise formulation of the thoughts/facts would substantially improve the readability and the scientific value of the paper. Therefore, I can recommend this manuscript for publication after some clarifications and corrections are made.

We thank the reviewers for the in-depth review which was helpful to improve the manuscript.

**General comments**

The measurement technique is heavily advertised as a new spectroscopic instrument. A total of 17 instances for "new instrument/generation spectrometers" can be found in the manuscript, which is clearly an exaggeration considering the fact that all of the key elements were already published by the authors almost a decade ago, i.e. in 2014/15. These include the OFFS-CRDS, the VCOF, the MZM, and even the Lamb dip approach for a highly accurate reference frequency measurements. As such, the "new" attribute should be avoided and used only in the context of the molecular species, i.e. $H_2O$, measurement. Therefore, I suggest to use an abbreviated name for the spectroscopic setup, e.g. VCOF-CRDS and use this throughout the manuscript.

It's true that it is best practice to avoid the use of "new" in scientific manuscript which are bound to age, and that the OFFS-CRDS and VCOF technique were introduced in (Burkart, 2015; Burkart et al., 2014) and the use of the lamb dip to determine frequency of absorption line in (Kassi et al., 2018). Using this specific spectral feature as a frequency reference has not been published anywhere else as far as we are aware. In principle, lamb dip self-referencing is not necessarily different from using the frequency of transitions in atomic clocks, but technically, in the context of a spectrometer that is used to monitor water vapour isotopic composition, including decreasing the pressure to saturate the molecular transitions, measuring the lamb, increasing back the pressure has not been published in any of our team's manuscripts.

In addition, none of the team of the manuscripts have been used to monitor water isotopes, and the instrument we describe here is actually fairly different than the one described in (Stoltmann et al., 2017), including the measurement under a constant flux of air, a mono-bloc VCOF (Casado et al., 2022), the self-frequency referencing.

We removed most of the instances of "new" instrument, leaving a single "new generation of instrument" in the introduction.

The key aspects relevant to the reader should be placed more prominently, e.g. in the abstract or conclusion: 1) the current instrument cannot be deployed in the field, 2) the lowest measured H$_2$O amount fractions are 25 ppm, while the performance at 1 ppm is merely an extrapolation, and 3) the recorded absorption spectrum is limited to 7 data-points only, which requires tedious frequency drift correction approaches.

We have modified the abstract and the conclusions to highlight some of these aspects.

We completely agree with aspect 1), and mentioned in the abstract a "laboratory-bound infrared spectrometer", and in the conclusion with the following sentence (lines 422 – 425):

> "While this instrument cannot be transported to Antarctica, by supporting measurement down to 1 ppmv of humidity, this technique shows great potential to finally be able to study the isotopic exchanges all year long in Antarctica where temperature can reach -80 °C during the polar night."

For aspect 2), we included that the performances at 1 ppmv are extrapolated, but also included addition evaluations of the precision down to 3.8 ppmv (see other comments answering Reviewer 1 as well as below) which support that the extrapolation is appropriate to justify the precision down to humidity levels of 1 ppmv, see the following sentence in the abstract:

> "This set up yields sub permil precision at water mixing ratios down to 4 ppmv, which suggest an extrapolated precision in δ18O of 1.5‰ at 1 ppmv"

For aspect 3), we respectfully disagree. We limited the number of points used for the absorption spectrum because the frequency of the laser is drifting so little that we do not need additional datapoints, not the other way around. Picarro instruments use the imbalance between the two points at the half-width of the transitions to evaluate how centred the central point is, here, this is not necessary due to the increased frequency stability of our laser source. Indeed, since the use of the optical feedback ensure that our spectral points are always systematically at the right frequency, we only need a single datapoint per absorption line to fully characterise the concentration in any of the water isotopes. To illustrate, the frequency drifts of the laser source shown in Figure 3, with excursions around 300kHz which are actively corrected, are one order of magnitude smaller than the frequency drift of a standard DFB laser diode (Casado et al., 2022), which are typically included in Picarro of OFCEAS instruments.

Especially, the validity of the extrapolation is questionable due to the spectral interferences from the CH$_4$ absorptions, the most prominent being at 7199.9547 cm$^{-1}$ (S = 3.715E-24) and at 7200.0287 cm$^{-1}$ (S = 1.742E-23), the former overlapping with H$_2$$^{18}$O absorption line, while the latter biasing the baseline at the 3$^{rd}$ data-point according to the Fig.2a. Without the discussion of these aspects and their impact on the atmospheric H$_2$O isotope measurements, the statements regarding the precision and accuracy at low amount fractions of H$_2$O remain highly speculative.

We acknowledge that the impacts of these methane absorption lines are not taken into account in our current manuscript, as the setup relied on synthetic air devoid of any methane. It will be necessary to indeed add a spectral point to also monitor the methane concentration. This has been mentioned in the discussion as an additional caveat before obtaining a field instrument:

> "Another caveat before the instrument can be deployed for field work is to take into account interferences from other species. For instance, taking into account the impact

of methane absorption features at 7199.95 cm$^{-1}$ and 7200.03 cm$^{-1}$ will require adding an extra spectral point for Antarctic field study as the methane absorption should be as strong as the water ones at humidity level around 1 ppmv."

This being said, adding an additional spectral point is absolutely doable while maintaining the repetition rate below 0.5 Hz, and the fairly monotonous methane concentration will not affect significantly our ability to monitor the isotopic composition similar to the approach in (Chaillot et al., 2023), so we do not believe that the precision and accuracy proposed here can be considered speculative.

There is a great deal presented related to drifts and their suppression. The authors successfully demonstrate the benefit of the Lamb dip method for frequency self-referencing. While the comparison with the standard optical frequency comb is convincing, the authors should describe the details of the measurements: i.e. that the instrument is operated in slow-pace measurement mode specifying the resolution and scanning time, quantifying the impact on the stability of the optical cavity when switching the gas pressure from 35 to 0.1 mbar, and the overall duty-cycle of the measurement.

This is a fair argument. Additional detail about the whole self-referencing process has been included in section 2.3:

> "Here, we make use of this property to evaluate the frequency drift of the laser source by scanning every hour the Lamb dip feature associated with the H$_2$$^{18}$O transition at 7199.96cm$^{-1}$. To do so, we decrease the pressure inside the cavity down to 0.1 mBar, let the cavity stabilise to this new experimental conditions for 2 minutes, and then measure Lamb dip feature across 4 MHz at extremely high resolution (30 kHz) for 7 minutes. This method provides with frequency measurements of the Lamb dip feature centre with a precision of 2.5 kHz in a single scan (Kassi et al., 2018). During a period of 140 hours, we measure the frequency deviation from the Lamb dip method, which reproduces exactly the ones estimated from the measurement of a beatnote with an optical frequency comb locked on a GPS (Burkart et al., 2014) (Fig. 3), with a correlation between the two times series of r² = 0.995 (p < 0.05). The pressure is then increased back to 35 mbar, within 30 seconds. This leads to an artificial isotopic composition measurement for another 3 minutes, after which, we can measure again the vapour isotopic composition. The overall duty-cycle here was around 13 minutes every hour during which the instrument was not able to monitor isotopic composition. Improvements on the VCOF laser source as recommended in (Jobert et al., 2022) could limit the need for such self-referencing cycles. Here, we used this measurement of the deviation of the frequency of the laser source as a self-referencing method which can be applied every hour to ensure that the deviation of the laser source frequency remains smaller than 10 kHz."

Also the sudden transition shown on Fig.3 around 90 h requires some discussion to better understand the reason and its effect on the retrieved precision.

We cannot explain the abrupt transition shown on Fig. 3, this is typical behaviour of the VCOF cavity as detailed also in (Casado et al., 2022). As for the effects on the retrieved precision, since we are actively correcting the frequency of the laser using either the lamb dip self-referencing or the optical frequency comb, they are exactly what is described in Figure 7. Such

frequency excursions are actively corrected during the use of the instrument. This has been explicitly mentioned in Section 3.3.:

> "The use of the lamb dip self-referencing approach enables to actively correct all frequency deviation of the VCOF laser source (Fig. 1a), leading to the performances displayed in Figure 7g)."

It's also worth noting that the frequency drift of a DFB laser which is not stabilised through optical feedback can experience drifts of several MHz within a few seconds (Casado et al., 2022). This is typically what is installed in a Picarro instrument, and a rudimentary wavemeter is used to take into account at least part of this drift. Here, the laser source is experiencing drift of 350kHz within a few hours. This looks impressive here because there is not point of reference, but this is neither a large nor a sudden transition.

Furthermore, it would be interesting to see the effectiveness of this approach in case of larger temperature variations, mimicking real field conditions.

As the VCOF laser source is temperature stabilised (Casado et al., 2022), the environmental temperature variations are irrelevant. Furthermore, when deployed in Antarctica, the instrument will not be outside but in a temperature controlled laboratory, and thus, not be under the influenced of temperature variations significantly larger than in the tests realised in this manuscript.

Many statements about limitations or issues are generalized; however, these are mainly specific to CRDS, or more strictly speaking to Picarro instruments. Benchmarking of the setup to a L2130i/ L2140i is fine, but then the discussion should be confined accordingly.

The instrument is also benchmarked against OFCEAS instruments (Landsberg et al., 2014). We have done extensive bibliographic research and these are the only instruments which are mentioned to measure water isotopic composition at low humidity for the last decade. For temperature and tropical regions, a few studies are still based on Los Gatos instruments, some of which are based on multipass cell, but this is extremely marginal, and usually the performances are not as good as Picarro instruments for low humidity conditions. We benchmarked our set-up against the most precise and recognised instrument in our field, and generalised these results to all commercial instruments, since Picarro instruments have been and still are the reference in term of water vapour isotopic composition monitoring. As a result, we believe that our statements are justifiably generalised to commercial instrument for water vapour isotopic composition monitoring. The following statement has been added to the introduction:

> "Throughout this manuscript, we will use Picarro instruments as a benchmark for commercial instruments considering how ubiquitous they are in water vapour isotopic composition monitoring."

**Specific comments**

The title needs some revisions. The "low humidity water vapour" is not appropriate. I suggest the "low amount fraction" instead. In general, the humidity should be replaced by amount fraction or mixing ratio throughout the paper, especially when is expressed in ppm.

We respectfully disagree. While it's true that humidity is a very generic term, it's also understandable by everyone. "Amount fraction" can refer to anything. Mixing ratio, is commonly used for water vapour content, but can also be used for any mixture. Overall, humidity is here used for *specific humidity* which is often expressed in ppm as well, and is extremely closed from the mixing ratio. It is widely used in the community (Berkelhammer et al., 2016; Noone et al., 2011; Oerter et al., 2019; Ricaud et al., 2014; Risi et al., 2012; Steen-Larsen et al., 2015). We feel like using *specific humidity* in this context in the title is going to create an unnecessarily complicated title but are open to modifying it further. The current version taking account the suggestions of the reviewer is:

> "Reliable low humidity water vapour isotopic composition measurements using frequency stabilised cavity ring down spectroscopy"

The term "new infrared instrument" is very general (see general comments) and thus, I recommend to be more specific here, e.g. "frequency stabilized CRDS".

We agree and modified the title accordingly.

In my opinion, the "reliable" is somewhat far-fetched and misleading, mainly because the low (< 20 ppm) $H_2O$ amount fractions were not measured directly (and extrapolation is not straightforward) and there was no demonstration of any calibration with V-SMOW standard materials or reference gas mixtures with different isotopic composition. The authors should either include such data or refrain from creating false expectations.

To answer the concerns that no measurements was done at humidity levels below 20 ppmv, additional measurements, down to 3.8 ppmv, were added. These datapoints were not included in the extrapolation but used as a validation and confirm the results of the extrapolation.

The extrapolation is actually straightforward since it is directly linked with the signal to noise ratio. The lower water vapour concentration, the less molecules, the lower signal to noise ratio. It is expected that the precision would follow a power law with the humidity. In addition, we actually show a power law behaviour over two orders of magnitude (Figure 3c), which demonstrates that this is an appropriate extrapolation.

We included additional measurements of six internal standards themselves referenced regularly on the SMOW/SLAP scale, see lines 284 to 302:

> "Another aspect of the accuracy of isotopic analysers is the linearity of the instrument, or its isotope-isotope response. We evaluated the precision of the new instrument on the V-SMOW/V-SLAP scale by measuring six internal standards from our institute ranging from –54 to +0.5‰. We used the water vapour generator (Leroy-Dos Santos et al., 2021) to generate stable humidity levels of 90 minutes and average out the last 15 minutes. As this water vapour generator is not as versatile as an automatic sampling device, injecting water with different isotopic composition (especially across a range of dozens of permil) is cumbersome due to extended memory effects in the humidity generator and it was necessary to wait more than 12 hours to do a new isotopic sample. The measured δ18O aligns perfectly with the internal standard values (r² virtually undifferentiable from 1, N

= 9), and the residuals of the linear regression have a standard deviation of 0.04‰ (Figure 7).

[Figure]

*Figure 7: Isotope-isotope response: measurements of internal standards referenced at average humidity of 800 to 1200 ppmv on the SMOW-SLAP scale and linear regression of the measured values against the standard values, compared to the residues.*

This demonstrates the linearity of the isotopic measurement of the VCOF-CRDS. Overall, isotopic monitoring via CRDS technique has been demonstrated to be extremely linear, even outside of the range of the isotopic compositions used for calibration (Casado et al., 2016; Steig et al., 2014). In practice, for isotopic composition measurements of ice core samples, it would be necessary to use the same sample preparation line (auto-sampler and vaporiser) for samples and calibration. If the residuals had the same type of distribution (standard deviation around 0.04 ‰), to reach an accuracy of 0.01‰ in $\delta 18 O$, it would be necessary to perform calibration on a smaller range of isotopic composition or to include more measurements of the standards."

In the abstract, the key findings should be mentioned: precision in d-values, integration time, $H_2O$ amount fraction, frequency stability, laser emission frequency, optical path length, SNR value, drift value, etc. The last sentence about the new constrains on the fitting technique requires more details in the manuscript and a clear statement in the abstract about the key factor that makes the biases less disturbing.

We included the precision in $\delta^{18}O$ and d-excess for a given H2O amount fraction and a given integration time. The frequency stability was already the topic of (Casado et al., 2022). The laser emission frequency is not relevant here, this could be done with other transition and would clutter the abstract. The optical path length does not apply to CRDS since the time that the light spends in the cavity is determined by the ring down time, and thus, vary when scanning inside a transition or the baseline. The SNR depends on the humidity levels at which we are measuring. The drift is already specifically mentioned.

The statement about the fitting technique has been amended.

The statement on Pg2, L45: "Applications of infrared spectroscopy techniques to trace detection and isotopic monitoring is dominated by two techniques: OF-CEAS and CRDS" is simply overlooking the scientific fact that the most accurate and field-deployed measurements are performed with the classical direct absorption spectroscopy (e.g. Aerodyne TILDAS and similar techniques) extending the capabilities even toward clumped isotopes.

As for the previous general comment, classical direct absorption spectroscopy techniques are rarely used for water isotopes. We introduced the following changes to restrict our introduction to water isotopes which is the topic of this manuscript:

> "Applications of infrared spectroscopy techniques to water isotopic monitoring is dominated by two techniques:"

TILDAS instruments are out of the framework of this manuscript since they are not used for water isotopes.

The authors should consider adding more details regarding their setup: the supplier of the DFB laser, optical power, operating parameters (current, voltage, and temperature), the optical power reaching the detector, detector type and vendor, the supplier of the CRD mirrors and the coating, piezo actuator, etc.

We included additional relevant details about the various components of the setup.

Please motivate the selection of the CRD cavity length: why 48 cm instead of the more optimal 23 cm?

We realised that the size of the cavity was not optimal while characterising it. A longer cavity is favourable to obtain a better signal to noise ratio. In order to rapidly *jump* from mode to mode, it is necessary to calculate the exact length that would enable to reach the top of each transition without actuating the piezo. This did not appear to be such a major point when we design the first cavity, and we are sharing this information so people can avoid repeating our mistake. The actual 23 cm length is specific to our instrument, and has been replaced by:

> "Changing the length of the CRDS cavity so the FSR aligns the frequency of the measurement at the top of all the transitions align with the centre of the feature would mitigate a large part of the drift impact on $\delta 18O$ (this is equivalent to finding a FSR equal to the common denominator between the difference of frequency between the three transitions)."

Without a detailed comparison between slow- and high-pace mode, it is rather difficult to understand why the high-pace mode is favoured in this work. Does the water vapour isotopic composition change faster than the time required for a full spectral scan and if even so, would this high temporal resolution be useful in any context? Contrary, there are many beneficial aspects for using the slow full spectrum mode, such as higher spectral point density, less prone to frequency drifts, accurate and robust fitting, etc. I do not really understand what is meant by changes in the gas composition during the slow scan and the additional noise with their auto-correlated features. These statements require more clarification about the underlying effect and processes.

This was our exact reasoning when we started to measure with the slow pace mode: having more spectral points might help. In the end, exactly because the water vapour (content and isotopic composition) change within a few seconds, having spectra that last 30 seconds to a minute lead to noise on the absorption and additional noise on the isotopic measurements. As frequency drift is absolutely not a problem for us due to the optical feedback (Casado et al., 2022), the slow full spectrum mode is not valuable. The following sentence to reinforce this point has been added to this section:

> "Fast variations of humidity level during a scan creates large uncertainty in the fitting procedure which leads to additional noise on the isotopic composition which are not averaged out due to their auto-correlated features. The high pace scans are associated with higher instantaneous noise with little auto-correlation which can be averaged out rapidly. As a result, when the high pace scans are averaged out to the resolution of the slow pace measurements, the precision of the multiple average high pace scans is better than the precision of single slow pace scans."

Fig.2 caption: by adding experimental parameters such as gas pressure, H2O amount fraction and optical path length would help the reader to better interpret the illustrated spectra.

We included the gas pressure and the mixing ratio in the caption. The optical path length is not relevant here in CRDS since it depends on the absorption as this is a resonant cell.

As the speed-dependent Nelkin-Ghatak profile is a function of seven parameters, it is not clear how this profile can be applied to a spectrum containing 7 data-points. Furthermore, it is not mentioned how the non-Voigt parameters were determined for the selected transitions.

This was indeed not very clear that the NG profile was not applied directly to our high pace spectra but rather fitted on the slow pace spectrum once, as a reference. Since the temperature and the pressure in the measurement cell are constant, we are able to fix most of the parameters of the NG profile. This has been specified (lines 144 to 150):

> "We fitted a slow pace/high precision spectrum using a speed-dependent Nelkin-Ghatak profile (SDNGP) (Long et al., 2011), similarly to the approach used to measure $CO_2$ isotopic composition (Stoltmann et al., 2017) to fix pressure and temperature dependant parameters. As temperature and pressure are constant, these parameters can be fixed which reduce the number of free parameters. We then were able to generate a simple, multi-linear, conversion matrix which could link the intensity of each transition in the spectra to the concentration of each isotope. "

It is shown that at low gas pressure the Lamb dip can be generated. While this is a very elegant and robust approach to determine absolute frequency, it leads to another question that was not mentioned in the paper, namely, the saturation effect also at higher gas pressure. It is well known that the field energy build-up in the cavity can be sufficiently high to induce non-linear effects and saturate the absorbing gas. How does this affect the accuracy of the isotope ratio measurements?

The saturation effect is indeed an important question for isotopic composition measurement, especially in high finesse cavity. As described in (Kassi et al., 2018), in the case of saturated absorption, the residues of the exponential fit due to the saturation can be clearly identified. In

our set up, we were able to reduce the power that we inject in the cavity to avoid saturation. We added the following sentence to the manuscript:

> "As mentioned above, the power injected to the cavity is amplified to 11 mW to increase the SNR on the photodiode but to ensure that saturation is not affecting the absorption profile of the gas inside the cavity (Kassi et al., 2018)."

Fig.4 shows the Allan-deviation (ADEV) plot over 2 days. This extended period is somewhat obsolete considering the optimal stability range of the spectrometer of 150 s, and also the fact that the authors do not disclose their gap-filling method to estimate the ADEV beyond the 10 h limit. On the other hand, the observed behaviour can fully be dominated by the water vapour generator. Thus, it remains an open question, why the authors did not use e.g. a pressurized air cylinder for the low $H_2O$ amount fraction measurements.

We disagree that having the ADEV plot over 2 days is obsolete. Yes, the optimal stability range is reached rapidly, but the fact that we reach a plateau and that the ADEV is not increasing with time scale is an indicator that there is no drift of the isotopic measurement at the scale of 2 days, which is due to our frequency stability in contrast for instance to a Picarro (Casado et al., 2016; Steig et al., 2014), or an OFCEAS instrument (Landsberg et al., 2014)

We are publishing on Github and Mathworks the gap-filling ADEV method (Matlab and Python versions are available) so they can be used in a more generalised setup.

We did not use pressurised dry air tank because water is an extremely sticky molecule and that without a dedicated prep line which has been pumped on and heated to more than 100°C for an extended period of time, we have not been able to successfully transfer dry air to the instrument. The instrument has not been open to outside air for the last two years, but at the 12 000 ppmv of the lab, every tube we have is completely saturated with water.

Nonetheless, in the new iteration of the manuscript, we actually took advantage of the desorption effects from a tube under a variable high flow of dry air (3 ppmv) to generate humidity levels below 20 ppmv. This has been added to Section 2.4:

> "2.4 Generation of water vapour standards
>
> We used the water vapour generator instrument described in (Leroy-Dos Santos et al., 2021) to generate stable water levels to evaluate the response of the infrared spectrometer to varying humidity levels. We used dry air bottles with less than 3 ppmv of water to supply the gas to the instrument. We used laboratory standards of water with isotopic composition varying from 0 to -60 ‰ to supply with water of known isotopic composition. The water vapour generator shows relatively good performances down to humidity around 20 ppmv where outgassing from the instrument itself limits its performance. As a result, we used to evaluate the precision as a function of humidity (section 3.1) and the humidity response of the instrument (section 3.2) down to 25 ppmv.
>
> To obtain an evaluation of the precision of the instrument below these humidity levels, we connected a dry air bottle directly to the instrument and regulated the produced humidity levels from outgassing of the tube by changing the flux of dry air to an exhaust connected to a sonic nozzle. Using this method, humidity levels down to 4 ppmv could be reached. In this case, the isotopic composition is not known so it cannot be used to

evaluate the accuracy of the instrument, only how stable the instrument is capable to measure relatively stable isotopic composition of outgassing water from the tube walls. As the temperature of the tube and the dry air canister were not regulated nor monitored, the resulting water vapour was relatively variable (variations around 10% of the produced humidity level), which lead to potential variability in the isotopic composition, leading itself to relatively high Allan standard deviations not necessarily linked with poor performance of the instrument."

And to the result section 3.1 and Figure 4:

"

[Figure]

***Figure 4:*** *Allan standard deviation plots of the δ18O measurements of the VCOF-CRDS instrument stabilised with an optical frequency comb: a) long-term Allan standard deviation plots realised at humidity levels of 25 and 400 ppmv with stable measurements of 7 days, b) 3D plot of the Allan standard deviation for different time scales and humidity levels, and c) evaluation of the precision of the instrument across humidity levels (dots) for measurements average over 1 second (blue), 2 minutes (red), as well as fit with a power law (solid line) and extrapolation from the fit at lower humidities that could not be reached with the calibration device (Leroy-Dos Santos et al., 2021);* *additional measurements between 3 and 10 ppmv obtained from outgassing tubes (section section 2.4) are included on the graph as upper limits of the precision below 25 ppmv; compared to typical commercial instrument behaviour (dashed grey line, linear approximation of the performances of a Picarro L2140i extracted from Fig. 3 of Leroy-Dos Santos et al, (2021) and of a Picarro L2130i from Casado et al, (2016)).*

The change of precision scales with humidity as shown in Figure 4c, and so at 1 Hz, the precision of the instrument at 30 ppmv is roughly 2 ‰ in δ18O (roughly 10 times larger since the humidity is approximately ten times smaller), dropping down to 0.1 ‰ after 800 seconds, and at 800 ppmv around 0.1 ‰ at 1 Hz and dropping to 0.005 ‰ after 800 seconds. While we were not able to generate stable moisture flux at 1 ppmv using the humidity generated described in (Leroy-Dos Santos et al., 2021), we extrapolated the precision expected after two minutes at humidity lower than 25 ppmv by fitting the data with a power law and find a precision of 1.5 ‰ at roughly 1 ppmv. We also directly connected the instrument to dry air bottle controlling the humidity from the outgassing of the tubes connecting the bottle to the instrument (see Methods, Section 2.4) and evaluated the precision at 3.8, 4.2, 6.5, and 11 ppmv. Since this method to generate water

standard is extremely dependent on temperature variations, it is only useful as an upper bound of the precision of the instrument as the humidity levels are relatively variable (standard deviation of the humidity larger than 10% of the humidity content), and thus, these datapoints where not included in the power law fit. We find precisions ranging from 0.5 to 0.7 ‰ after two minutes (Fig. 4c) which agree relatively well with the power law defined at larger humidity levels."

The term d-excess needs to be defined and it would be beneficial to also include the δD data. Eventually, move Appendix B into the main text by replacing Fig.4 by Fig.B1.

We defined the d-excess in the introduction: "and d-excess (second order parameter d-exc = δD - 8 δ$^{18}$O)."

For including δD, we respectfully disagree, the appendix B is only marginally important as we still show that δD behaves extremely similarly to δ$^{18}$O. The choice of only using δ$^{18}$O was already explained in the main text (lines 222 to 225 of the updated manuscript), both δ$^{18}$O and δD are extremely correlated and it would only clutter the manuscript.

Pg9, L201: The biases mentioned here are in the context of using the Picarro instruments and therefore generalization to any infrared spectrometer should be avoided.

Picarro instruments are amongst the most used instrument to monitor water isotopic composition. They clearly are the benchmark in this field, and for now, still have the highest performances (*en part* with OFCEAS instruments from AP2E, the manuscript about these instruments is being written by members of our institute). We included the following statement in the introduction (lines 43-44):

"Throughout this manuscript, we will use Picarro instruments as a benchmark for commercial instruments considering how ubiquitous they are in water vapour isotopic composition monitoring."

More information about the new fit parameters are required. How are they obtained, what exactly is optimized to reduce the biases and how much improvement is obtained compared to the case with improved frequency stabilization scheme alone?

The first part of the question has been answered on the question of the Speed-dependent Neklin-Ghatak. One further element is that the residue reduction function that we used to fit complete spectra could not, with reasonable computational power being required, optimise the fit to the sensitivity of the instrument of the order of $10^{-12}$ cm$^{-1}$. The uncertainties on the fit were, as a result larger than what was dictated by our precision. By fixing most parameters and using a relatively simple, multi-linear, conversion matrix between the intensity of the transitions and the isotopic composition.

On the last part of the comment, we agree that probably the improved frequency stabilisation scheme is the leading element for the improved precision, and probably to the improved humidity dependency. We removed the references to the "new fitting technique" in the abstract.

I recommend a revision of the Discussion section. Instead of recalling the limitations described in earlier publications (this can easily be moved to the introduction), the authors should keep

the focus on their own results and present a concise evaluation of the measurements, perhaps by answering all the above questions and addressing the raised issues.

We prefer to maintain the context in the discussion to make it easier for the readers.

We included elements about the cost of the set up compared to a commercial instrument, the impact of the humidity generator compared to our instrument, as well as perturbation from other species (with a specific emphasis on the methane transitions suggested by the reviewer) in a revised version of the discussion.

For sake of scientific rigour, Fig.8 should be slightly modified: the grey bar corresponding to the OFFS-CRDS performance should be drawn to the 20 ppm level, while the extrapolation toward 1 ppm should be indicated by a lighter gray or a dashed contour.

We included extra datapoints at humidity levels down to 3.8 ppmv which confirm the validity of the extrapolation (see previous comments). We believe that the Figure 8 is appropriate to show qualitatively the range of humidity at which the instrument can be used, without adding levels of complexity.

The main factors hindering the instrument to be deployed in the field are not fully clear. In particular, the issue caused by the thick ice layer is hard to understand.

On top of ice shelfs, there is no electrical ground because ice is an electric insulant. This really appears straightforward that in this case, having a dedicated electronics that can withstand static electricity shocks is of the utmost importance. This was already written in the main text: "instruments must be extremely isolated for static shocks due to the lack of electric ground with the thick ice layer)."

Furthermore, there is no attempt by the authors to discuss the potential improvements or alternative solutions to overcome this serious limitation. What are the constrains that can eventually be removed by further engineering and is there any fundamental issue that is difficult to address?

There are no technological issues to make a smaller, field deployable version of the instrument as far as we are aware now. When we can make a field version, we will. So far, the constrains are funding availability.

As a side note, a similar technology (see e.g. the ProCeas from AP2E) has been proven to be market ready.

We are aware and are developing these instruments with the company AP2E. A manuscript will be submitted shortly for peer-review but the instruments are not at the level of the performances we show here.

Without addressing these aspects, the final statement in the Conclusion remains highly elusive. Nevertheless, I'm firmly convinced that the scientific value of the manuscript is appropriate and with an adequate revision the objections can be completely overcome.

**Technical corrections**

Abstract: Pg.1,L12: add "water" before vapour

Taken into account.

Abstract: Pg.1,L16: replace "suffers" with "exhibits" or "demonstrates"

Taken into account.

Pg.2, L56: define VCOF-CRDS

Taken into account.

Pg.4, L75: replace "Distributed FeedBack diode (DFB)" with "distributed feedback (DFB) diode laser"

We would rather keep the capitalised letters to explain the acronym.

Pg.4, L91: check grammar "cavity can be ever so slightly adjusted"

Taken into account

Pg.4, L93: change "Bronkhorst pressure and flow controllers" to "pressure and flow controllers (Type/Model, Bronkhorst)"

Taken into account

Pg.4, L94: use SI units, i.e. Pa instead of mbar. Check all instances.

For the sake of simplicity, we believe that the metric system unit mbar is more relevant here than the use of Pa (30 mbar vs 3000 Pa).

Pg.4, L96: check grammar: "The instrument is set so the laser source produces …"

Taken into account

Pg.4, L100-104: consolidate the phrase

Unable to answer.

Pg.7, L148-150: revise the phrase, especially the "the accuracy of the instrument of humidity to isotope relationship" is difficult to interpret.

Taken into account

Pg.7, L158: check wording "water level stable levels lasting"

Taken into account

Pg.7, L166: change "a normal law" to "white-noise"

Taken into account

Pg.8, L190: replace "no" with "not"

Taken into account

Pg.9, L210: add space between value and unit. Check for all other instances.

Taken into account

Pg.10, Fig.6 caption: check "local tap distilled tap water"

Taken into account

Pg.13, L268: what is "fractionation coefficient"?

A reference has been added.

Pg.14, L301: replace "phrase" with "phase"

Taken into account

Pg.15, L310: rephrase "outside of the confine of a fully equipped spectroscopy lab", e.g. outside of temperature controlled laboratory environment.

The temperature control is not the main factor here because all our instruments are temperature regulated. Having access to a wavemeter, an optical frequency comb, makes measurement in a spectroscopy lab much easier than in the field.

**Reviewer #3:**

This manuscript reports on the development and characterization of a laser-based spectrometer for measurements of isotopic composition of water at low humidity levels, relevant for future field measurements in central Antarctica. This work fits very well within the scope of the AMT journal. The cavity ringdown spectrometer is based on a DFB laser whose frequency is locked by optical feedback to a temperature-stabilized reference cavity. The long-term frequency stability of the laser is ensured by monitoring a Lamb dip signal in a selected transition at regular intervals. The authors test and discuss the performance of the spectrometer for measurements of $\delta^{18}O$ and d-excess at various levels of humidity between 20-1500 ppm.

I have several major comments on this manuscript that I believe the authors should address before the paper is published.

We would like to thank the reviewer for his attentive evaluation of our manuscripts.

The paper claims that the spectrometer is able to measure isotopic composition down to (or even below) 1 ppmv water mixing ratios, conditions found in central Antarctica. I see two problems with this statement. First of all, the lowest water mixing ratio used in this work is 25 ppmv, and the result at 1 ppmv is an extrapolation. What is the rationale for extrapolating to 1 ppmv? One could extrapolate even further and discuss the expected performance at around 0.1 ppmv.

In the updated version of the manuscript, we included measurements at water levels of 3.8, 4.2, 6.5 and 10 ppmv to validate the extrapolation below 25 ppmv. The method to generate the humidity levels was different, so we did not include these datapoints in the linear regression, but more as a validation of the extrapolation below 10 ppmv to justify an extrapolation of more than one order of magnitude. We limited ourselves to 1 ppmv because this is the presumed lower limit of humidity levels found in Central Antarctica were temperature rarely go below -80°C.

Changes to the manuscript include in Section 2.4:

"2.4 Generation of water vapour standards

We used the water vapour generator instrument described in (Leroy-Dos Santos et al., 2021) to generate stable water levels to evaluate the response of the infrared spectrometer to varying humidity levels. We used dry air bottles with less than 3 ppmv of water to supply the gas to the instrument. We used laboratory standards of water with isotopic composition varying from 0 to -60 ‰ to supply with water of known isotopic composition. The water vapour generator shows relatively good performances down to humidity around 20 ppmv where outgassing from the instrument itself limits its performance. As a result, we used to evaluate the precision as a function of humidity (section 3.1) and the humidity response of the instrument (section 3.2) down to 25 ppmv.

To obtain an evaluation of the precision of the instrument below these humidity levels, we connected a dry air bottle directly to the instrument and regulated the produced humidity levels from outgassing of the tube by changing the flux of dry air to an exhaust connected to a sonic nozzle. Using this method, humidity levels down to 4 ppmv could be reached. In this case, the isotopic composition is not known so it cannot be used to

evaluate the accuracy of the instrument, only how stable the instrument is capable to measure relatively stable isotopic composition of outgassing water from the tube walls. As the temperature of the tube and the dry air canister were not regulated nor monitored, the resulting water vapour was relatively variable (variations around 10% of the produced humidity level), which lead to potential variability in the isotopic composition, leading itself to relatively high Allan standard deviations not necessarily linked with poor performance of the instrument."

And to the result section 3.1 and Figure 4:

"

[Figure]

***Figure 4:*** *Allan standard deviation plots of the $\delta^{18}O$ measurements of the VCOF-CRDS instrument stabilised with an optical frequency comb: a) long-term Allan standard deviation plots realised at humidity levels of 25 and 400 ppmv with stable measurements of 7 days, b) 3D plot of the Allan standard deviation for different time scales and humidity levels, and c) evaluation of the precision of the instrument across humidity levels (dots) for measurements average over 1 second (blue), 2 minutes (red), as well as fit with a power law (solid line) and extrapolation from the fit at lower humidities that could not be reached with the calibration device (Leroy-Dos Santos et al., 2021); additional measurements between 3 and 10 ppmv obtained from outgassing tubes (section section 2.4) are included on the graph as upper limits of the precision below 25 ppmv; compared to typical commercial instrument behaviour (dashed grey line, linear approximation of the performances of a Picarro L2140i extracted from Fig. 3 of Leroy-Dos Santos et al, (2021) and of a Picarro L2130i from Casado et al, (2016)).*

The change of precision scales with humidity as shown in Figure 4c, and so at 1 Hz, the precision of the instrument at 30 ppmv is roughly 2 ‰ in $\delta^{18}O$ (roughly 10 times larger since the humidity is approximately ten times smaller), dropping down to 0.1 ‰ after 800 seconds, and at 800 ppmv around 0.1 ‰ at 1 Hz and dropping to 0.005 ‰ after 800 seconds. While we were not able to generate stable moisture flux at 1 ppmv using the humidity generated described in (Leroy-Dos Santos et al., 2021), we extrapolated the precision expected after two minutes at humidity lower than 25 ppmv by fitting the data with a power law and find a precision of 1.5 ‰ at roughly 1 ppmv. We also directly connected the instrument to dry air bottle controlling the humidity from the outgassing of the tubes connecting the bottle to the instrument (see Methods, Section 2.4) and evaluated the precision at 3.8, 4.2, 6.5, and 11 ppmv. Since this method to generate water

Second, the measurements are performed at room temperature in the laboratory, and the discussion about how the instrument will be prepared for operation at -80 C is missing. On the contrary, it is stated that the instrument in its current form is not ready for field deployment, e.g., because of the fragile design of the reference cavity (line 293). Will it be possible to construct a stable reference cavity, which is the heart of the system, that will operate at -80 C? In fact, the temperature stabilization of the presented spectrometer is not discussed in the manuscript. The authors should add this information, and comment on future developments that will enable field deployment.

Similar to the deployment of commercial instruments in the field (Bréant et al., 2019; Casado et al., 2016), the instrument would not be deployed outside, but a heated inlet is used to bring outside air in the gas analyser. So the instrument will never have to perform at -80°C. The temperature stabilisation of the laser source is described in (Casado et al., 2022; Jobert et al., 2022) in great length. We included the following statements to describe the temperature stabilisation of both the laser source and the gas analyser (lines 93-94, and 111-113, respectively):

"The entire laser source set-up is heated up at 26°C by heating elements and PT1000 temperature sensors as described in (Casado et al., 2022; Jobert et al., 2022)."

"The entire gas analyser (Fig. 1g) is stabilised using a peltier device (Supercool) at a temperature of 28°C. The CRDS cavity is stabilised using a heating wire and PT1000 temperature sensors at 29.000°C."

I suggest the authors modify the phrasing at a couple of places to better reflect the focus of this work. For example, in the Abstract the second sentence states: 'During polar winter, the temperatures are so low that current commercial techniques are not able to measure the vapour isotopic composition with enough precision' which puts focus on measurements at low temperatures, which are not shown in this work. I suggest focusing on low humidity levels, and stating clearly that the instrument was tested down to 25 ppmv. Similarly, in the Discussion, it is stated that 'the new instrument can circumvent *all the hurdles* that limit the monitoring of water vapour isotopic composition monitoring in the coldest conditions, such as found in Antarctica in winter' (line 288), and that 'The performances of the instrument in the field should be roughly the same, provided that the temperature stabilisation of the instrument is as good as in the AC rooms of the lab (line 296)', which are far-fetched claims, since it is not even discussed how the temperature was stabilized in the laboratory.

While this is technically true, the water partial pressure is completely dominated by the temperature in this case. We suggest the following updated sentence in the abstract:

"During polar winter, the temperatures, and thus the specific humidity, are so low that current commercial techniques are not able to measure the vapour isotopic composition with enough precision."

For the temperature stabilisation, see the previous comment, since our instruments are temperature controlled, the impact of moderate temperature variations do not influence our measurement capabilities.

The authors state that one of the keys to the good performance of the instrument are 'new constraints on the fitting technique' (line 17) and 'the new fit parameters' (line 207), but the details of the fitting technique are missing. It is also not clear what the authors are comparing to when saying 'new'. On line 117, the authors state that the absorption profile is fitted using a speed-dependent Nelkin-Ghatak profile, but it is not clear how and why this is done, since only one point is recorded per absorption line. The fits and residuals are not shown, it is not stated what the fitting parameters are, and if any parameters are fixed – what are they fixed to? Where are the line intensities, frequencies and broadening parameters taken from?

This was indeed not very clear that the NG profile was not applied directly to our high pace spectra but rather fitted on the slow pace spectrum once, as a reference. Since the temperature and the pressure in the measurement cell are constant, we are able to fix most of the parameters of the NG profile. This has been specified (lines 138 to 143):

"We fitted a slow pace/high precision spectrum using a speed-dependent Nelkin-Ghatak profile (SDNGP) (Long et al., 2011), similarly to the approach used to measure $CO_2$ isotopic composition (Stoltmann et al., 2017) to fix pressure and temperature dependant parameters. As temperature and pressure are constants, these parameters can be fixed which reduce the number of free parameters. We then were able to generate a simple, multi-linear, conversion matrix which could link the intensity of each transition in the spectra to the concentration of each isotope. The typical standard deviation of the fit residuals is around $2 \times 10^{-12} cm^{-1}$, leading to a signal to noise ratio of the order of 10 000."

Additionally, we removed the reference to the fitting technique in the abstract, as, as noted by Reviewer 2, the impact on the isotope-isotope and humidity-isotope response of the frequency stabilisation cannot be separated from our fitting technique.

The authors refer to the $2 \times 10^{-12}$ $cm^{-1}$ noise level in the residuals, but they do not show these residuals. Does this number refer to the noise in the fit to the 7 points shown in Fig. 2a? On line 281 it is stated that the performance of the previous systems was limited by drifting fringes. How were the fringes avoided in this system? Is there no baseline in these measurements? The authors should show and describe the fits, including the residuals.

The noise level is obtained when fitting the slow pace datasets, for which individual datapoints are obtained with the same number of ring down. The fits and the complete spectra have been included in multiple past manuscripts (Burkart et al., 2014; Stoltmann et al., 2017), as this manuscript was meant to describe the performances in term of isotopic measurement, and not of absorption, we decided to focus on the data with the additional level of abstraction. We none the less agree that it is confusing to include this sentence about the residual at this point of the manuscript, as it's more relevant for the slow pace measurement. This sentence has been removed.

The fringes are mentioned to describe the OFCEAS instrument, which is a completely different technology based on the difference of incoming and outgoing signal from a resonant cavity to determine the absorption feature. This method has been known to be very sensitive to fringes (Romanini et al., 1997). Here, we are using CRDS in the gas analyser, and since we are measuring the decay time of the light inside the resonant cavity, fringes are not as much of an issue as for OFCEAS (Burkart et al., 2014).

The baseline is measured by 4 of the spectral points as shown in Figure 2a).

Related to the fitting process, please explain how the measurement points for the high pace mode were selected. The laser frequency is tuned in steps of cavity FSR, how is it ensured that some points coincide with the line centers? Is this a lucky coincidence, or was the FSR chosen carefully? Why would a cavity length of 23 cm be better than 48 cm (line 263)? That would make the FSR larger, which would make hitting specific frequencies harder. What is the 'parking method' mentioned in the figure caption on line 113?

The measurement points were selected to hit as much as possible the top of the transitions, as already described in the manuscript (lines 141 and 142 of the updated version): "In the high pace measurement mode, the spectra are composed of 7 datapoints (one at the top of each transition, and four for the baseline, Fig. 2a)".

The points actually do not coincide with the line centres which is the exact topic of Section 3.4 as this affects a lot of the drift of the instrument. The 23cm is not necessarily important for a general case. We just calculated that for these transitions with our specific set up, it would be an acceptable solution where we could find a common denominator between the FSR and the frequency difference between the three transitions. We modified the text accordingly (lines 350 to 354):

> "Changing the length of the CRDS cavity so the FSR aligns the frequency of the measurement at the top of all the transitions align with the centre of the feature would mitigate a large part of the drift impact on δ18O (this is equivalent to finding a FSR equal to the common denominator between the difference of frequency between the three transitions)."

The term "parking" has been removed from the caption.

Minor comments

- Line 53: 'we present a new generation of infrared spectrometers' – I suggest toning this down to 'we present a new spectrometer'.

Well, the bottom line is that we plan to build more instruments to measure not only water isotopes but other species as well, that would be built on this set up, so it's not just a new instrument.

- Line 100: 'with performances of $10^{-12}$ cm$^{-1}$' – 'performances' should be 'sensitivity', and the time over which this sensitivity is obtained should be stated.

Taken into account.

- I suggest unifying the name of the technique, it is called OFFS-CRDS or VCOF-CRDS at different places.

Unfortunately, the technique *OFFS-CRDS* was the original term coined by (Burkart et al., 2014) while later on the term *VCOF-CRDS* instrument was introduced (Stoltmann et al., 2017). We tried to keep OFFS for the technique et VCOF for the instrument throughout the manuscript.

- Line 105: 'spectral resolution is of only multiple of the FSR' – this is sample point spacing, not resolution.

The sentence has been modified.

- Lind 135: I suggest removing references in the caption of figure 3, because they suggest that the data are taken from these papers, while they are not.

Taken into account.

- Line 164: 'at the resolution of the instrument (3 Hz)' – It is not clear where this resolution comes from. Is this time or frequency resolution? I guess it is the latter, because 0.7 s does not correspond to 3 Hz. Please clarify.

It was the sampling rate from the instrument which was increased while we wrote the manuscript (as it's directly controlled by the number of ring downs per spectral point we take). We corrected the information in the method section (lines 141-142):

"In the high pace measurement mode, the spectra are composed of 7 datapoints (one at the top of each transition, and four for the baseline, Fig. 2a) and are realised in approximately 0.3 s when averaging together 10 ring downs per spectral point."

- Line 160: State how was the Allan deviation modified.

The modified Allan standard deviation routine (coded in both Python and Matlab) will be uploaded on Github and Mathworks and made freely available with this manuscript.

- Line 189: 'roughly 10 times larger' – larger than what? Further on – I do not understand where it is shown in Fig. 4c what happens after 800 s ('dropping down to … after 800 s').

This has been modified.

- How was the isotope-humidity response measured? What does anomaly mean here? Is it the same as in Fig. 2b?

We included a sentence to explain how the humidity response is obtained (lines 259 to 261):

"To evaluate the humidity response of the VCOF CRDS, we used the water vapour generator to generate different humidity levels and evaluate how the measured isotopic composition deviate at low humidity from the expected value for the given standard."

The reference that was used to calculate the anomaly was included in the caption of Figure 6.

- Line 224: 'The main source of uncertainty of the measurement comes from the drift of the laser source' – Is it frequency or intensity drift?

Taken into account.

- The language needs to be carefully revised at many places, for example:
  - Line 16: the instrument exhibits (not suffers)
  - Line 47 and 106: the interest of – the advantage?
  - Line 96: stable light – what is stable about the light?
  - Line 101: monitor water vapour isotopic (what?)
  - Lines 148-150 are hard to follow
  - Line 190: while we were not able to …. generated as described
  - Line 267-269 are incomprehensible
  - Line 300 is incomprehensible

Taken into account.

**Bibliography:**

Berkelhammer, M., Noone, D. C., Steen-Larsen, H. C., Bailey, A., Cox, C. J., O'Neill, M. S., Schneider, D., Steffen, K. and White, J. W. C.: Surface-atmosphere decoupling limits accumulation at Summit, Greenland, Sci. Adv., 2(4), 2016.

Bréant, C., Dos Santos, C. L., Agosta, C., Casado, M., Fourré, E., Goursaud, S., Masson-Delmotte, V., Favier, V., Cattani, O. and Prié, F.: Coastal water vapor isotopic composition driven by katabatic wind variability in summer at Dumont d'Urville, coastal East Antarctica, Earth Planet. Sci. Lett., 514, 37–47, 2019.

Burkart, J.: Optical feedback frequency-stabilized cavity ring-down spectroscopy - Highly coherent near-infrared laser sources and metrological applications in molecular absorption spectroscopy, University Grenoble Alpes., 2015.

Burkart, J., Romanini, D. and Kassi, S.: Optical feedback frequency stabilized cavity ring-down spectroscopy, Opt. Lett., 39(16), 4695–4698, doi:10.1364/OL.39.004695, 2014.

Casado, M., Landais, A., Masson-Delmotte, V., Genthon, C., Kerstel, E., Kassi, S., Arnaud, L., Picard, G., Prie, F., Cattani, O., Steen-Larsen, H. C., Vignon, E. and Cermak, P.: Continuous measurements of isotopic composition of water vapour on the East Antarctic Plateau, Atmos. Chem. Phys., 16(13), 8521–8538, doi:10.5194/acp-16-8521-2016, 2016.

Casado, M., Stoltmann, T., Landais, A., Jobert, N., Daëron, M., Prié, F. and Kassi, S.: High stability in near-infrared spectroscopy: part 1, adapting clock techniques to optical feedback, Appl. Phys. B, 128(3), 1–7, 2022.

Chaillot, J., Dasari, S., Fleurbaey, H., Daeron, M., Savarino, J. and Kassi, S.: High-precision laser spectroscopy of H2S for simultaneous probing of multiple-sulfur isotopes, Environ. Sci. Adv., 2023.

Jobert, N., Casado, M. and Kassi, S.: High stability in near-infrared spectroscopy: part 2, optomechanical analysis of an optical contacted V-shaped cavity, Appl. Phys. B, 128(3), 56, doi:10.1007/s00340-022-07779-x, 2022.

Kassi, S., Stoltmann, T., Casado, M., Daëron, M. and Campargue, A.: Lamb dip CRDS of highly saturated transitions of water near 1.4 μm, J. Chem. Phys., 148(5), 54201, doi:10.1063/1.5010957, 2018.

Landsberg, J., Romanini, D. and Kerstel, E.: Very high finesse optical-feedback cavity-enhanced absorption spectrometer for low concentration water vapor isotope analyses, Opt. Lett., 39(7), 1795–1798, doi:10.1364/OL.39.001795, 2014.

Leroy-Dos Santos, C., Casado, M., Prié, F., Jossoud, O., Kerstel, E., Farradèche, M., Kassi, S., Fourré, E. and Landais, A.: A dedicated robust instrument for water vapor generation at low humidity for use with a laser water isotope analyzer in cold and dry polar regions, Atmos. Meas. Tech., 14(4), 2907–2918, doi:10.5194/amt-14-2907-2021, 2021.

Long, D. A., Bielska, K., Lisak, D., Havey, D. K., Okumura, M., Miller, C. E. and Hodges, J. T.: The air-broadened, near-infrared CO2 line shape in the spectrally isolated regime: Evidence of simultaneous Dicke narrowing and speed dependence, J. Chem. Phys., 135(6), 2011.

Noone, D., Galewsky, J., Sharp, Z. D., Worden, J., Barnes, J., Baer, D., Bailey, A., Brown, D. P., Christensen, L. and Crosson, E.: Properties of air mass mixing and humidity in the subtropics from measurements of the D/H isotope ratio of water vapor at the Mauna Loa

Observatory, J. Geophys. Res. Atmos., 116(D22), 2011.

Oerter, E. J., Singleton, M., Thaw, M. and Davisson, M. L.: Water vapor exposure chamber for constant humidity and hydrogen and oxygen stable isotope composition, Rapid Commun. Mass Spectrom., 33(1), 89–96, 2019.

Ricaud, P., Carminati, F., Courcoux, Y., Pellegrini, A., Attié, J.-L., El Amraoui, L., Abida, R., Genthon, C., August, T. and Warner, J.: Statistical analyses and correlation between tropospheric temperature and humidity at Dome C, Antarctica, Antarct. Sci., 26(03), 290–308, 2014.

Risi, C., Noone, D., Worden, J., Frankenberg, C., Stiller, G., Kiefer, M., Funke, B., Walker, K., Bernath, P., Schneider, M., Bony, S., Lee, J., Brown, D. and Sturm, C.: Process-evaluation of tropospheric humidity simulated by general circulation models using water vapor isotopic observations: 2. Using isotopic diagnostics to understand the mid and upper tropospheric moist bias in the tropics and subtropics, J. Geophys. Res. Atmos., 117(D5), D05304, doi:10.1029/2011JD016623, 2012.

Romanini, D., Kachanov, A. A., Sadeghi, N. and Stoeckel, F.: CW cavity ring down spectroscopy, Chem. Phys. Lett., 264(3–4), 316–322, doi:http://dx.doi.org/10.1016/S0009-2614(96)01351-6, 1997.

Steen-Larsen, H. C., Sveinbjörnsdottir, A. E., Jonsson, T., Ritter, F., Bonne, J., Masson-Delmotte, V., Sodemann, H., Blunier, T., Dahl-Jensen, D. and Vinther, B. M.: Moisture sources and synoptic to seasonal variability of North Atlantic water vapor isotopic composition, J. Geophys. Res. Atmos., 120(12), 5757–5774, 2015.

Steig, E. J., Gkinis, V., Schauer, A. J., Schoenemann, S. W., Samek, K., Hoffnagle, J., Dennis, K. J. and Tan, S. M.: Calibrated high-precision 17O-excess measurements using cavity ring-down spectroscopy with laser-current-tuned cavity resonance, Atmos. Meas. Tech., 7(8), 2421–2435, doi:10.5194/amt-7-2421-2014, 2014.

Stoltmann, T., Casado, M., Daëron, M., Landais, A. and Kassi, S.: Direct, Precise Measurements of Isotopologue Abundance Ratios in CO2 Using Molecular Absorption Spectroscopy: Application to Δ17O, Anal. Chem., 89(19), 10129–10132, 2017.

---

## Author Response (AR2)

Dear editor,

We would like to thank you and the reviewer 3 for their comments. We have been working toward a new version of the manuscript taking their respective comments into account. We include the comments from the reviewers in black, our responses in blue, and the modifications to the manuscript in red in this response file.

All the best,

On the behalf of all the co-authors,

Mathieu Casado

**Reviewer #3:**

I commend the authors on the form that their manuscript has taken.

All of my comments and concerns have been addressed with the supplied revisions.

There still exist a few grammatical inconsistencies (I will not list them here), however they do not hinder the communication of the scientific concepts and can easily be addressed in the copy editing process.

I look forward to following the future development of this new generation of stable water isotope analyzer.

We thank the reviewer for his comments that helped improve the manuscript. We'll work with the copy editing team to remove the final grammatical inconsistencies.

**Reviewer #5:**

The clarity of the manuscript is greatly improved, I thank the authors for their efforts.

We thank the reviewer for their comments which improved the manuscript, and we will implement the following comments.

I have a couple of minor comments:

The sensitivity on line 118 should be stated together with the corresponding measurement time.

We added the corresponding measurement time (Line 118):

"with a sensitivity of $10^{-12}\ cm^{-1}$ after 60 s."

Line 152: the Lamb dip frequency is not a physical constant; it is affected by pressure and power shifts. Rephrase.

This is technically correct. We have modified the text as follow:

"their frequency, which remains constant for constant conditions of pressure, can be used as a frequency reference."

The use of the lamb dip here is justified by the low values of line pressure shift, about 0.01 cm⁻¹/atm, i.e. 300 MHz/atm, or an absolute shift of 30 kHz @ 0.1 mBar. With reproducible conditions of pressure for each Lamb dip scan (better than 0.01mBar), the precision of the pressure shift is known around 2.5 kHz. We do not mention the power shift, which has not been observed (AC Stark shift). Note than again, the power is kept constant because the ring downs are triggered by a constant value of photodiode power.

Line 234 and on: I still do not understand where one can see what happens after 800 s in Fig. 4c. The longest averaging time there is 2 min, which is 120 s, not 800 s.

This was not strictly linked to Figure 4c but to Figure 4b and 4a. This has been added to the text.

The following paragraph is still incomprehensible:

Line 190: 'We separate the performances of the instrument with first, the precision and the drift of the instrument in the long term, 190 secondthe accuracy of the instrument, includingthe humidity to isotope and isotope to isotope relationships, and finally highlight the results of the frequency auto-referencing on the performances of the instrument.'

We modified the paragraph as follow:

"We discuss the performances of the instrument with first, the precision and the long-term drift of the instrument; second, the accuracy of the instrument, including the humidity-to-isotope and isotope-to-isotope relationships; and finally by highlighting the impacts of the frequency auto-referencing on the performances of the instrument."

The language is still imprecise at places, and there are some typos:

Line 16: Technics –> techniques

Taken into account

Line 55: 'to generate very precise absorption spectroscopy' –> What does 'generating spectroscopy' mean?

Replaced by "perform"

Line 105: 'the power injected to the cavity is amplified to 11 mW to increase the SNR on the photodiode but to ensure that saturation is not affecting the absorption profile of the gas inside the cavity' –> is 'but' correctly used here? Change to 'while ensuring'?

Taken into account

Line 109: Peltier

Taken into account

Line 120: 'realising spectra' –> measuring?

Taken into account

Line 131, caption of Fig 2(a): please keep the nomenclature consistent, high pace – slow pace, or high resolution?

Taken into account

Line 165: It is not clear what 'aritificial isotopic composition measurement' means. What is artificial?

Replaced with "biased"

Line 195: what is meant by 'performances'?

Modified to: "To evaluate the drift in measured isotopic composition associated with the technique directly,"

Line 199: 'generate water as stable as possible' –> Do you mean 'stable water concentration'? Water itself is stable.

Taken into account

Line 240: were, not where

Taken into account

Line 283: what does 'average out the last 15 minutes' mean? Was this data used or not?

Modified to: "and included the average value of the last 15 minutes".

Fig. 6: a and b are missing in panels

Taken into account

Line 310: 'distance of the spectroscopic measurement to the centre of the absorption feature' –> detuning?

It can be explained like that, but we already used "detuning" twice before in the same paragraph to discuss when the distance was artificially increased, and so, we prefer this formulation.

Line 322 Lamb dip (also in fig. 8g)

Taken into account

Line 340: Changing the length of the CRDS cavity so the FSR aligns the frequency of the measurement at the top of all the transitions 340 align with the centre of the feature would mitigate a large part of the drift' –> 'the frequency of the measurement' should be the sampling points, and there are too many 'align'in this sentence

Taken into account

Line 435: Comparaison -> comparison

Taken into account

Wrong punctuation: line 19, 63

Taken into account

Plural/singular mixed: line 66, 125, 126, 160, 198, 251, 316, 361

Taken into account

**Reviewer #4:**

In the revised manuscript, most of the concerns raised by the referees were addressed and implemented leading to an overall improvement of the paper. Therefore, I can recommend this manuscript for publication after some minor corrections are made.

We thank the reviewer for their effort providing advice on our manuscript which greatly improved its quality.

General comments:

The precision of the instrument seems to be limited by the water vapor generation system, because the spectrometer is operated in flow-through (25 sccm) regime. Thus, fluctuations associated to evaporation, desorption effects, etc. contribute to the short and long-term instabilities. Despite of these issues, the authors do not consider the batch mode, i.e. enclosing the gas sample into the optical cavity and measuring the isotope ratio on such static samples. What is the reason for not doing so?

Water is an extremely sticky molecule, the current cavity was electropolished, but no further hydrophobic treatment was set up. In these conditions, the exchanges between the surface of the cavity and the atmosphere would dominate in batch mode. In principle, we could wait long enough that everything is at equilibrium. Yet, the water vapour monitoring that is required for ice core analysis or in-situ field study are always done under flow-through regimes, in part to reduce the exchange with the cavity, even in commercial instruments where hydrophobic silicosteel treatments are applied. Obtaining the performances of the set-up in static (batch mode) regime would not be representative of what could be measured in realistic conditions. Finally, the performances we are currently able to obtain are already one order of magnitude better than commercial instruments.

As the instrument is a laboratory setup, the (Antarctic) air samples have to be transferred to the instrument. A flow-through scheme would obviously means a large gas consumption. The authors should shortly discuss how they will address this issue.

There are several potential uses for this instrument. The current instrument is a proof of concept that will remain in a laboratory. In the lab, it can be used to measure ice core samples, which are routinely measured with a flow-through scheme because of how sticky water is. In this case, to generate 25sccm of air with a humidity level appropriate for measurements requires less than 0.1mL of melted ice. We currently run a dozen commercial analysers working on this principle in our institute alone. For these applications, the water and dry air consumptions are not an issue.

We are currently building the new generation of instrument that will go the field. The goal has never been to bring back canister of Antarctic air back to the institute to be measured. For these applications, we hope that very soon the instruments will be deployed in Antarctica, and as such, the gas consumption will not be an issue.

The spectral interference by CH4, especially at low humidity levels, is not addressed in a satisfactory manner. Just stating that the problem can be simply solved by adding another spectral point does not work, because the spectral separation of the transitions is not equidistant and by no means coincides with the FSR of the cavity. Therefore, the authors should either quantify the expected impact on the accuracy of the H2O isotope ratio or admit that their

reported precision and accuracy only holds for water vapor in synthetic (N2 and O2 mixture) air.

We respectfully disagree. This has already been done in (Chaillot et al., 2023) with a similar experimental setup dedicated to H2S with satisfactory performances. As we are monitoring the exact same transitions that Picarro instruments where the methane concentration is actively fitted using spectral points that are only focusing on the water transitions and not adding dedicated spectral points to aim specifically at the top of the methane transition. We believe that it is a reasonable assumption to estimate that we can do the same by adding spectral points to evaluate the methane concentration. We added a reference to Chaillot et al in the main text:

> "For instance, taking into account the impact of methane absorption features at 7199.95 cm-1 and 7200.03 cm-1 will require adding an extra spectral point for Antarctic field study as the methane absorption should be as strong as the water ones at humidity level around 1 ppmv, following a similar approach than (Chaillot et al., 2023)."

Also, the FSR of the current instrument is 297 MHz, the typical width of a methane transition at 35 mbar is several GHz, so it appears also perfectly reasonable that we would be able to aim at or near the summit of the transition while maintaining a sampling scheme based on the FSR, but the instrument is also able to actively adjust the length of the cavity as detailed in the method, even though it is slower. We could also envision scanning at a different frequency than FSR resolution if absolutely necessary. We believe that this discussion is out of the framework of this manuscript.

Specific comments:

Title: By definition, humidity is the amount of water vapour (the gaseous form of water) in the air. Saying "humidity water vapor" is exceedingly redundant. I suggest: "Reliable water vapour isotopic composition measurements at low humidity using frequency stabilised cavity ring down spectroscopy"

This is a very good point, we modified the title accordingly.

Abstract: "We produced a laboratory-bound infrared spectrometer…" change to "We present a laboratory infrared spectrometer…"

Taken into account

Abstract: "based on all these technics dedicated to measure…" change to "leveraging on these techniques to measure…"

Taken into account

Abstract: "The instrument is additionally not hindered…" change to " The instrument is not hindered… "

Taken into account

Pg1, L17: the "water stable isotope" was correct, the "stable water isotope" is not.

Taken into account

Pg2, L65: replace "in" with "at" Antarctic conditions

Taken into account

Pg3, L71: remove "for laboratory and field monitoring of"

Taken into account

Pg4, L94: the heating is only due to the heating elements, while the PT1000 is to monitor the temperature. Please check phrasing.

Taken into account

Pg5, L114: same here.

We respectfully disagree, we indicated that the cavity is "stabilised", and the sensors are needed to actively control the temperature.

There are at least 3 different temperatures in the system: the analyser is stabilized at 28°C, the laser source setup at 26°C, while the CRDS cavity at 29°C. What is the reason of not keeping everything at the same temperature?

The CRDS cavity is heated specifically 1°C higher than the analyser to be able to regulate the temperature using only a heating element, because it is much simpler. The laser source is for now external and the temperature was suggested by Schott as good for the Zerodur to have low expansion.

Pg4, L108-110: Please rewrite, e.g.: "As mentioned above, the power injected into the cavity is amplified to 11 mW to increase the SNR on the photodiode, while making sure that no saturation is taking place that may affect the absorption profile of the gas inside the cavity."

Taken into account

Pg5, L126: replace "must be" with "are"

Taken into account

Pg5, L128: add value for the scan duration

Taken into account

Pg5, L129: quantify the far slower

Taken into account

Pg5, L131: "noise on the isotopic composition" should read "fluctuation in the isotopic composition"

Taken into account

Pg5, L133: remove "out"

Taken into account

Pg5, L134: replace "resolution" with "acquisition time"

Taken into account

Pg5, L134: delete "multiple"

Taken into account

Pg8, L176: what do you mean by "artificial isotopic composition measurement"?

Modified to "biased"

Pg8, L177: revise this sentence. If the period is one hour and the system is not measuring for 13 min then the duty-cycle is about 80 %.

Modified to: "The overall duty-cycle here was around 80%, including 13 minutes every hour during which the instrument was not able to monitor isotopic composition"

Pg8, L183: replace "stable water levels" by "stable humidity levels"

Taken into account

Pg8, L184: what is a dry air bottle? Do you mean pressurized-air or synthetic-air cylinder?

Taken into account

Pg9, L210: replace "successive water vapour stable humidity levels" with "successive stable humidity levels"

Taken into account

Pg10, L245: "Reciproquely" is not an English word.

Taken into account

Pg10, L246: replace "Hz" with "s"

Taken into account

Pg12, Fig6: the "several" is actually only two: high and low.

Taken into account

Pg13, L295: "evaluated the precision", most likely you mean accuracy.

Correct, thank you.

Pg13, L297: delete "out" or replace with "for"

Taken into account

Pg16, L355-359: remove this paragraph, because it is highly speculative and practically cannot be achieved. Also the sentence is more confusing than explaining.

We respectfully disagree. This is not speculative; we are currently building a cavity that is 25cm long. We included this so it can help other people who would want to build similar setups, to indicate that the targeted transitions (and their respective distance) should be considered when conceiving things as trivial as the exact length of the measurement cavity.

Pg17, L376: replace "here and based" with "here is based"

Taken into account

Pg17, L382: replace "Relying on relatively cheap, fibered lasers which are commonly build for telecommunication, …" with " Relying on cost effective, fibered telecom lasers"

Taken into account

Pg18, L395: replace "be needed to disentangle the variability coming from the water generation from the one coming from the measurement." with " be needed to disentangle the various contributions"

Taken into account